# MULTI-VIEW ADAPTIVE PARTITIONING WITH GLOBAL ASSOCIATION FOR GRAPH ANOMALY DETECTION

## ABSTRACT

Graph anomaly detection has emerged as a fundamental technique for detecting anomalies in complex relational data across diverse domains. A key challenge in graph anomaly detection is the difficulty deep models face in representing tabular node attributes. Unlike image and text data, whose samples usually reside on a single, smooth, and differentiable manifold, tabular samples scatter across fragmented manifolds lacking manifold connectedness. This fragmentation violates the local-smoothness assumption that most deep networks rely on, leading to degraded performance. To address the above challenge, a Multi-view Adaptive Partition Encoder (MAPE) is proposed. Multiple complementary adaptive partition operators are introduced in MAPE to discretize the feature space and assign learnable embeddings to the resulting subspaces, thereby reducing manifold connectedness. Furthermore, sharing a sub-space is treated as evidence of high-order affinity between nodes, forming the basis of the proposed Multi-Pattern Global Association (MPGA) module for capturing global dependencies. Extensive experiments across 10 benchmarks demonstrate that the proposed method consistently outperforms 27 competitive baselines, including recent state-of-the-art models.

## 1 INTRODUCTION

The rapid growth in scale and complexity of contemporary data has established *graph anomaly detection* (GAD) as a crucial technique across many domains. By modeling relational structures, GAD identifies suspicious financial transactions (Pourhabibi et al., 2020), facilitates early detection of medical anomalies (Wang et al., 2020), and enhances productivity and reliability in smart manufacturing (Wu et al., 2021). These capabilities mitigate risks and substantially improve system safety and efficiency, highlighting an urgent need for advanced GAD methods.

Two persistent challenges remain in graph anomaly detection: (1) tabular node attributes are mixed-type, high-dimensional, and semantically heterogeneous, hindering expressive representation learning, and (2) graph topology is non-Euclidean with long-range, multi-hop, multi-scale dependencies, whereas conventional methods based on local message passing or a single long-range mechanism have finite receptive fields and capture only limited long-range patterns, thus failing to model global structural dependencies on large graphs.

The fundamental challenge of modeling tabular data with deep neural networks can be better understood by analyzing their manifold distributions. Unlike image and text data, whose samples usually reside on a single, smooth, and differentiable manifold, tabular samples scatter across fragmented manifolds lacking manifold connectedness (as illustrated in Figure 1). This fragmentation violates the local-smoothness assumption that most deep networks rely on, leading to degraded performance. Moreover, even when mapped via deep neural networks such as MLP (Rosenblatt, 1958) or decision-tree-based models such as XGBoost (Chen & Guestrin, 2016), tabular data still fail to restore manifold connectedness (as illustrated in Figure 2). Hence, an embedding strategy capable of mitigating manifold fragmentation of tabular data is crucial.

A Multi-view Adaptive Partition Encoder (MAPE) is introduced to mitigate manifold fragmentation in node features. Several complementary adaptive-partition operators are instantiated, each learning an independent rule to divide the raw feature space into a set of subspaces. All samples falling into the same subspace are mapped to a shared, trainable, view-specific embedding, and concatenating the embeddings from multiple operators yields a discrete-semantic node representation.

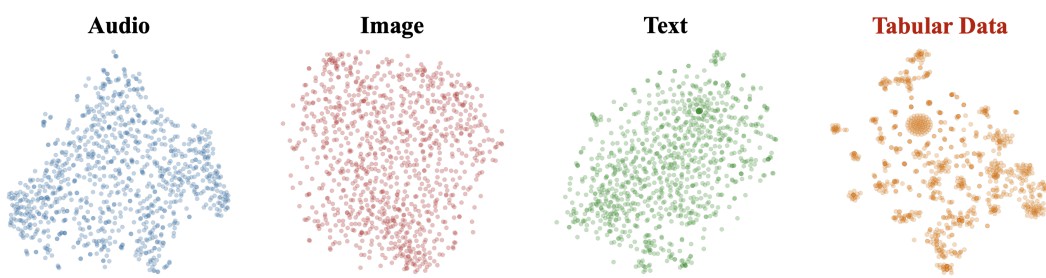

Figure 1: **Manifold connectedness across data modalities.** t-SNE projections of raw feature vectors are shown for audio (Warden, 2018), images (Krizhevsky et al., 2009),text (Lang, 1995), and tabular data (Kumar et al., 2019). Perceptual data (audio, image, text) lie on a single, smooth manifold, preserving coherent Euclidean neighbourhoods and local-smoothness assumptions. By contrast, tabular data break into numerous isolated micro-clusters separated by large voids, lacking the stable local structure needed for dependable neighbourhood relations.

This rescaling and re-balancing decomposes over-dense clusters and enlarges under-represented regions, thereby harmonizing sample density across the feature manifold and effectively establishing its manifold connectedness (as shown in Figure 2).

Building on the subspace assignments provided by MAPE, the Multi-Pattern Global Association (MPGA) module captures global dependencies from two complementary sources. First, representation-based association is computed from cross-view co-assignment statistics, reflecting how frequently two nodes share subspaces across views. Second, behavior patterns are constructed by per-view histograms of in- and out-neighbor assignments, providing a discrete characterization of neighborhood behavior. These association signals are integrated to supply a global affinity structure that efficiently models long-range, multi-scale dependencies on large, noisy graphs. In summary, the resulting Multi-view Adaptive Partitioning with Global Association (MAPGA) framework induces manifold connectedness in tabular data while enabling effective global-dependency modeling.

Grounded in manifold connectedness, this work explains why tabular node attributes attached to graph nodes in GNN-based anomaly detection are difficult to model and introduces a MAPE to induce manifold connectedness together with a MPGA module for multi-pattern global-dependency modeling. Across 10 public datasets, MAPGA outperforms 27 competitive baselines, including recent state-of-the-art methods. Ablation studies further confirm the complementary contributions of the discrete-partition encoder (MAPE) and the global-dependency module (MPGA).

## 2 RELATED WORK

Existing graph anomaly detection (GAD) research can be reorganized along two complementary axes: *node feature embedding for tabular attributes* and *global relation modeling*.

### 2.1 NODE-ATTRIBUTE REPRESENTATIONS

This line of work focuses on mapping numeric/discrete node attributes into discriminative representations that stabilize downstream relational learning. Classical tabular learners—including the Multi-Layer Perceptron (MLP) (Rosenblatt, 1958), k-Nearest Neighbors (KNN) (Cover & Hart, 1967), Support Vector Machine (SVM) (Chang & Lin, 2011), and Random Forest (RF) (Breiman, 2001)—offer nonlinear mappings from raw features. Gradient-boosted variants such as XG-Boost (Chen & Guestrin, 2016) and its outlier-oriented extension XGBOD (Zhao & Hryniewicki, 2018) further enhance detection accuracy on tabular data by enriching feature interactions.

Beyond pure tabular models, several approaches explicitly tailor embeddings to graph settings. BGNN (Ivanov & Prokhorenkova, 2021) couples GBDT with GNN training so that tree-based gradient updates refine node embeddings. For heterogeneous graphs, RGCN (Schlichtkrull et al., 2018) and HGT (Hu et al., 2020) introduce type-dependent parameters to project node/edge attributes into

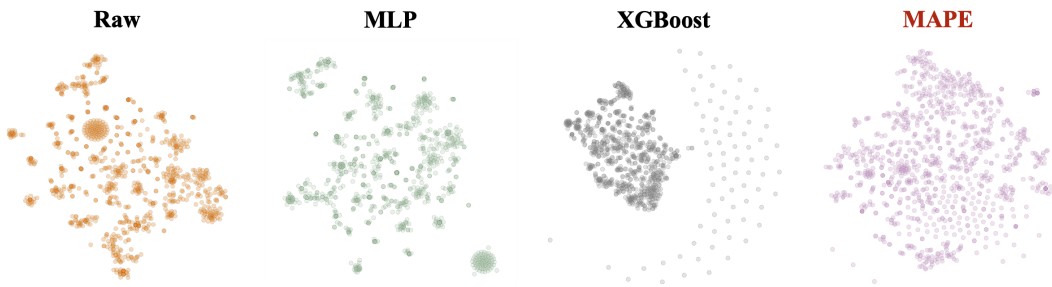

Figure 2: **Impact of different encoders on manifold connectivity of tabular data.** t-SNE projections of (a) the raw tabular features and of the representation spaces obtained by three embedding methods—(b) a two-layer MLP, (c) XGBoost leaf-index encoding, and (d) MAPE. A shallow MLP slightly contracts intra-cluster variance yet leaves most clusters isolated, while XGBoost collapses a few dense regions but pushes isolated outliers even farther apart. In contrast, MAPE bridges formerly disconnected clusters and yields a single, well-connected manifold with smooth density transitions, reducing the local-smoothness prior required by downstream neural models.

aligned subspaces; GIN (Xu et al., 2018) increases expressiveness for distinguishing subtle local patterns, which benefits initial node encodings. Self- and semi-supervised objectives also improve embeddings: DCI (Wang et al., 2021) decouples representation learning from labels via clustering-style pretext tasks; GGAD (Qiao et al., 2024) fabricates pseudo-outliers to enlarge discriminative margins; ADA-GAD (He et al., 2024) employs anomaly-denoised pretraining with a distribution regularizer; and ConsisGAD (Chen et al., 2024b) adds homophily-aware consistency through learnable augmentations. Combined pipelines from GAD-Bench (Tang et al., 2023) (RFGraph, XGBGraph) integrate tree ensembles with GNN embeddings to exploit both tabular modeling and relational signals. Recent hybrids further strengthen the attribute side: DGA-GNN (Duan et al., 2024) encodes non-additive attributes via decision-tree (binning) vectors and leverages feedback-driven grouping before hierarchical GNN aggregation; GAAPA (Duan et al., 2025) partitions attribute spaces into fine-grained bins and fuses the resulting attribute-pattern features with neighbor messages.

**Limitations.** Current node–feature encoders for tabular data primarily optimize discriminative risk or compression, yet they do not explicitly *establish manifold connectedness* in the representation space. Without manifold connectedness, the local-smoothness prior can lead to ill-conditioned optimization dynamics, which hampers stable convergence to good optima, especially under class imbalance and scarce anomalies. These limitations call for embedding mechanisms that *induce manifold connectedness* while preserving discrete semantics, thereby enabling reliable training of downstream deep networks.

## 2.2 GLOBAL DEPENDENCY MODELING

This line aims to encode multi-hop and distant relations while remaining robust under class imbalance and scarce anomalies. Message-passing backbones such as GCN (Kipf & Welling, 2016), SGC (Wu et al., 2019), GraphSAGE (Hamilton et al., 2017), GIN (Xu et al., 2018), and GAT (Veličković et al., 2017) expand the receptive field by aggregating neighbors layer-by-layer. More expressive architectures—Graph Transformer (GT) (Shi et al., 2020), PNA (Corso et al., 2020), BGNN (Ivanov & Prokhorenkova, 2021), RGCN (Schlichtkrull et al., 2018), and HGT (Hu et al., 2020)—incorporate global attention, multi-aggregator designs, and type-aware parameters to adapt to irregular connectivity and heterogeneous relations.

Specialized GAD models further tailor the relational pipeline to noisy links, heterophily, and camouflaged behaviors. GAS (Li et al., 2019) adapts to heterophilic/heterogeneous settings for robust propagation. PC-GNN (Liu et al., 2021) combines label-balanced sampling with a learnable distance function to filter misleading neighbors. GAT-sep (Zhu et al., 2020) separates ego and neighbor channels and incorporates higher-order neighborhoods to improve discrimination under heterophily. Spectrally stronger GNNs such as BernNet (He et al., 2021) and AMNet (Chai et al., 2022) increase capacity for multi-scale mixing, while GHRN (Gao et al., 2023) reduces cross-class interference via

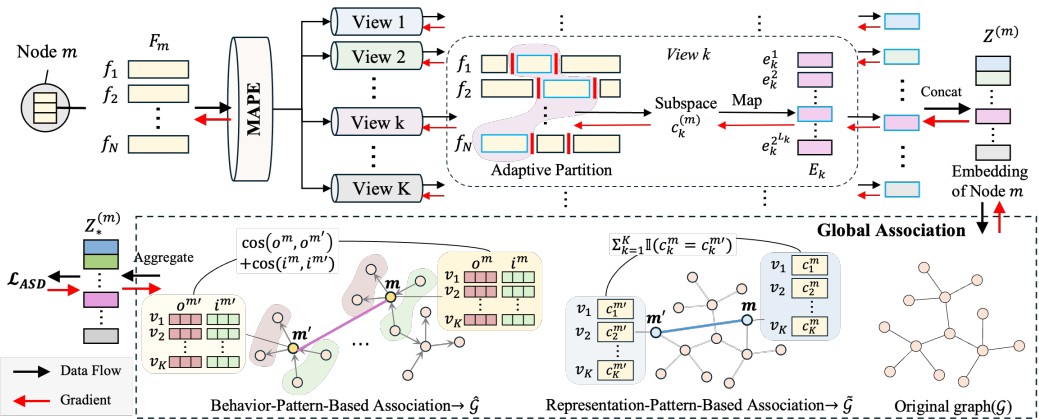

Figure 3: Overall architecture of MAPGA. MAPE partitions the representation space into $K$ learned views and maps each per-view subspace assignment to a view-specific embedding; concatenating the $K$ embeddings forms a discrete semantic node representation. MPGA builds representation-based affinity via cross-view co-assignment and models behavior patterns with per-view histograms of in- and out-neighbor assignments; the two are sparsified and fused with the adjacency to produce $\mathbf{A}^\star$, supplying long-range, pattern-aware context to the anomaly-detection backbone.

edge pruning. For multi-relational graphs, CARE-GNN (Dou et al., 2020) and H2-FDetector (Shi et al., 2022) explicitly handle mixed homophilic and heterophilic interactions. Semi-supervised and training-light advances include GGAD (Qiao et al., 2024), ADA-GAD (He et al., 2024), and ConsisGAD (Chen et al., 2024b), which inject pseudo-outliers, denoised pretraining, and consistency constraints into the relational pipeline; TFGAD (Zhou et al., 2025) provides a training-free scorer via low-rank reconstruction and projection errors. Recent unification efforts such as GADAM (Chen et al., 2024a) and UniGAD (Lin et al., 2024) integrate local–global cues via adaptive message passing and multi-scale readout sampling. Hybrid systems also contribute on the relational axis: GAD-Bench (Tang et al., 2023) couples ensembles with GNN embeddings; DGA-GNN (Duan et al., 2024) and GAAPA (Duan et al., 2025) combine attribute partitioning with hierarchical or neighbor-aware aggregation to capture global attribute–association patterns.

**Limitations.** Prevailing approaches to global dependency modeling rely on a single similarity-based relational mechanism defined in one representation space. Such feature-space–dependent single-relation designs have been observed to under-represent heterogeneous or low-similarity but semantically relevant associations, leaving certain long-range dependencies under-exploited (Meng et al., 2024).

FIX

**Summary.** Existing research on *Node feature embedding* and *global dependency modeling* provides essential building blocks for graph anomaly detection, but two gaps remain. First, most node–feature encoders for tabular data optimize discriminative objectives or compression without *explicitly inducing manifold connectedness* in the representation space, which violates the local-smoothness prior of gradient-based neural networks and impedes stable convergence under class imbalance and sparse anomalies. Second, prevailing global dependency modeling typically relies on a *single relational mechanism* applied to the *original node representations*, limiting the capacity to capture latent long-range dependencies and under-utilizing global structure. Accordingly, this work introduces MAPGA, which reconstructs manifold connectivity for tabular node attributes, preserves discrete semantics, and captures global associations across diverse patterns within a single framework; a brief theoretical analysis is provided in Appendix A.7.

## 3 METHOD

This section first presents the *Multi-view Adaptive Partition Encoder* (MAPE), which discretizes the feature space via learnable partitioners and maps each per-view subspace assignment to a view-

specific embedding, yielding a discrete-semantic node representation. It then introduces the *Multi-Pattern Global Association* (MPGA) module, which derives representation-based and behavior-based association graphs from these per-view assignments and fuses them with the original adjacency to form a global affinity graph for subsequent aggregation and anomaly discrimination. The overall framework is illustrated in Figure 3.

### 3.1 MULTI-VIEW ADAPTIVE PARTITION ENCODER

Let a node[1] $m$ be represented by a feature vector $\mathbf{F}^{(m)} = \{f_1^{(m)}, f_2^{(m)}, \ldots, f_N^{(m)}\} \in \mathbb{R}^N$. To overcome the manifold fragmentation effect analysed in Figure 1, the proposed framework first decomposes the feature space through a set of *learnable* linear partition operators that act in multiple, complementary *views*. Formally, we instantiate a collection of $K$ views $\mathcal{V} = \{v_1, v_2, \ldots, v_K\}$. Within a given view $v_k$ we define $L_k$ binary **decision rules** as follows:

$$\mathcal{R}_k = \{r_k^1, r_k^2, \ldots, r_k^{L_k}\}, \qquad r_k^\ell = (n_k^\ell, s_k^\ell),$$

where * $n_k^\ell \in \{1, \ldots, N\}$ indexes the feature $f_{n_k^\ell}^{(m)}$ selected by the $\ell$-th rule in view $k$; * $s_k^\ell \in \mathbb{R}$ is a *learnable split point* initialised uniformly and updated by back-propagation.

For node $m$, the output of rule $r_k^\ell$ is computed via a Heaviside step[2]:

$$b_{m,k}^\ell = H(f_{n_k^\ell}^{(m)} - s_k^\ell) \in \{0, 1\}. \tag{1}$$

Stacking the $L_k$ binary outcomes yields an $L_k$-dimensional code $\mathbf{c}_k^{(m)}$ as follows:

$$\mathbf{c}_k^{(m)} = [b_{m,k}^1, b_{m,k}^2, \ldots, b_{m,k}^{L_k}]^\top \in \{0, 1\}^{L_k}. \tag{2}$$

Geometrically, Eq. 1–2 realise an axis-aligned hyper-rectangular partition of the subspace defined by the features $\{f_{n_k^\ell}\}_{\ell=1}^{L_k}$. Each unique binary code specifies a *discrete sub-space cell* (partition region). Because all split points $s_k^\ell$ are trainable, the partition adapts to data distribution, thereby restoring local coherence even across fragmented manifolds.

For each view $v_k \in \mathcal{V}$ a shallow decision tree $T_k$ is first trained on the raw feature space using the Gini impurity criterion. Every internal split node of $T_k$ is converted into one binary decision rule. Consequently, the number of rules in this view equals the number of split nodes, $L_k = |\text{Splits}(T_k)|$. Let node $\ell$ split on feature index $n_k^\ell$ at threshold $t_k^\ell$. The corresponding rule parameters are initialised as $r_k^\ell = (n_k^\ell, s_k^\ell = t_k^\ell)$, after which all thresholds $s_k^\ell$ become *learnable* and are jointly updated with the downstream task loss (see Eq. 1).

The rule outcomes for node $m$ in view $k$ form the binary code $\mathbf{c}_k^{(m)} \in \{0, 1\}^{L_k}$ defined in Eq. 2. Because there exist $2^{L_k}$ distinct codes, we allocate a view-specific trainable dictionary $\mathbf{E}_k \in \mathbb{R}^{2^{L_k} \times d_0}$, where $d_0$ is a hyper-parameter controlling the embedding dimensionality of each view. The mapping function $\phi_k : \{0, 1\}^{L_k} \to \mathbb{R}^{d_k}$ is implemented by table lookup:

$$\mathbf{z}_k^{(m)} = \phi_k(\mathbf{c}_k^{(m)}) = \mathbf{E}_k[\text{idx}(\mathbf{c}_k^{(m)})], \tag{3}$$

where the integer index $\text{idx}(\mathbf{c}) = \sum_{\ell=1}^{L_k} c^\ell 2^{\ell-1}$ enumerates all possible codes in little-endian order. Both $\mathbf{E}_k$ and the split points $\{s_k^\ell\}$ are optimised end-to-end via back-propagation; no additional supervision is required.

Each view contributes one embedding: $\mathbf{z}_k^{(m)} \in \mathbb{R}^{d_0}$. The final representation of node $m$ is the concatenation

$$\mathbf{z}^{(m)} = [\mathbf{z}_1^{(m)} \| \mathbf{z}_2^{(m)} \| \cdots \| \mathbf{z}_K^{(m)}] \in \mathbb{R}^d, \quad d = \sum_{k=1}^K d_0, \tag{4}$$

which is fed to subsequent graph-aware networks (GNN, transformer, or MLP), thereby injecting *learnable, multi-granular, and globally comparable* discrete semantics into the downstream anomaly-detection pipeline (see Appendix A.2 and Appendix A.7–A.8 for further analysis).

---

[1] For clarity, the terms *node*, *sample*, and *instance* are used interchangeably.

[2] A smooth surrogate such as $\sigma(\alpha(f_{n_k^\ell}^{(m)} - s_k^\ell))$ with slope $\alpha$ is adopted during optimisation to preserve differentiability; the hard threshold is recovered at inference by $\text{round}(\cdot)$.

## 3.2 MULTI-PATTERN GLOBAL ASSOCIATION

The discrete partitions produced by the MDPE encoder endow every node with $K$ complementary observations that can be exploited to recover *long-range* dependencies unreachable by purely local message passing. We therefore construct three interrelated graphs and merge them into a unified structure $\mathcal{G}^\star$ that drives the subsequent aggregation.

**Notations.** Let the original directed graph be $\mathcal{G} = (\mathcal{V}, \mathcal{E})$, and recall the view-wise one-hot code $\mathbf{h}_k^{(m)} \in \{0,1\}^{2^{L_k}}$ whose non-zero index equals $\mathrm{idx}\big(\mathbf{c}_k^{(m)}\big)$. For compactness we denote by $\langle \mathbf{a}, \mathbf{b} \rangle$ the dot product $\mathbf{a}^\top \mathbf{b}$.

**Representation Pattern Graph $\widetilde{\mathcal{G}}$.** Two nodes are considered semantically close when they are assigned to the *same* discrete sub-space in many views. Formally, in view $k$ the coincidence indicator is $\langle \mathbf{h}_k^{(m)}, \mathbf{h}_k^{(m')} \rangle \in \{0,1\}$. Averaging over all views yields the Common-Subspace Frequency,

$$\phi_{mm'} = \frac{1}{K} \sum_{k=1}^K \langle \mathbf{h}_k^{(m)}, \mathbf{h}_k^{(m')} \rangle, \quad \phi_{mm'} \in [0,1]. \tag{5}$$

Setting edge weights $\widetilde{w}_{mm'} = \phi_{mm'}$ defines the representation pattern graph $\widetilde{\mathcal{G}} = (\mathcal{V}, \widetilde{\mathcal{E}})$.

**Behaviour-Pattern Graph $\widehat{\mathcal{G}}$.** While $\widetilde{\mathcal{G}}$ captures *node-level* co-occurrence, structural anomalies may manifest in *behavioural* discrepancies between neighbourhoods. Because each one-hot vector $\mathbf{h}_k^{(n)}$ is an indicator of its sub-space, summation over neighbours produces a histogram whose entries count how many neighbours fall in each sub-space.

For node $m$ and view $k$, let $N_{\text{out}}(\cdot)$ and $N_{\text{in}}(\cdot)$ denote the out- and in-neighbor sets, then

$$\mathbf{o}_k^{(m)} = \sum_{n \in \mathcal{N}_{\text{out}}(m)} \mathbf{h}_k^{(n)}, \qquad \mathbf{i}_k^{(m)} = \sum_{n \in \mathcal{N}_{\text{in}}(m)} \mathbf{h}_k^{(n)}, \tag{6}$$

followed by $\ell_1$-normalisation to obtain $\hat{\mathbf{o}}_k^{(m)}, \hat{\mathbf{i}}_k^{(m)} \in [0,1]^{2^{L_k}}$. Concatenating all $K$ views yields the *out-* and *in-behaviour* descriptors $\hat{\mathbf{o}}^{(m)}, \hat{\mathbf{i}}^{(m)} \in [0,1]^{\sum_k 2^{L_k}}$.

A symmetric cosine kernel measures behavioural affinity:

$$\psi_{mm'} = \cos\big(\hat{\mathbf{o}}^{(m)}, \hat{\mathbf{o}}^{(m')}\big) + \cos\big(\hat{\mathbf{i}}^{(m)}, \hat{\mathbf{i}}^{(m')}\big), \psi_{mm'} \in [0,2]. \tag{7}$$

Setting edge weights $\widehat{w}_{mm'} = \psi_{mm'}$ defines the behaviour-pattern graph $\widehat{\mathcal{G}} = (\mathcal{V}, \widehat{\mathcal{E}})$.

**Unified Graph Representation $\mathcal{G}^\star$.** The final affinity matrix is the convex combination:

$$\mathbf{A}^\star = \lambda_1 \mathbf{A} + \lambda_2 \widetilde{\mathbf{A}} + \lambda_3 \widehat{\mathbf{A}}, \quad \lambda_1, \lambda_2, \lambda_3 \geq 0, \sum_{i=1}^3 \lambda_i = 1, \tag{8}$$

where $\mathbf{A}, \widetilde{\mathbf{A}}, \widehat{\mathbf{A}}$ are the row-normalised adjacency matrices of $\mathcal{G}, \widetilde{\mathcal{G}}, \widehat{\mathcal{G}}$, respectively, and $\{\lambda_i\}$ are tuneable hyper-parameters that balance local and global cues.

### 3.2.1 NEIGHBOURHOOD AGGREGATION OVER $\mathcal{G}^\star$

The unified affinity matrix $\mathbf{A}^\star$ (Eq. 8) encodes *local*, *representation-level*, and *behaviour-level* relations simultaneously. To fully exploit these global couplings, an $L$-layer Graph Convolutional Network (GCN) is applied directly on $\mathcal{G}^\star$.

Let $\mathbf{Z}^{(0)} = [\mathbf{z}^{(1)}, \ldots, \mathbf{z}^{(|\mathcal{V}|)}]^\top \in \mathbb{R}^{|\mathcal{V}| \times d}$ be the concatenated node embeddings, and let $\widetilde{\mathbf{A}}^\star = \mathbf{D}^{-\frac{1}{2}} \mathbf{A}^\star \mathbf{D}^{-\frac{1}{2}}$ denote the symmetrically normalised adjacency. An $L$-layer GCN produces

$$\mathbf{Z}^{(L)} = \mathrm{GCN}_\theta\big(\widetilde{\mathbf{A}}^\star, \mathbf{Z}^{(0)}\big), \quad \mathbf{z}_\star^{(m)} = \big[\mathbf{Z}^{(L)}\big]_{m,:},$$

followed by a linear head that outputs class probabilities. Training uses the standard binary cross-entropy ($\mathcal{L}_{ASD}$). Since our contributions lie in the node feature embedding and the multi-view global dependency modeling that define $\mathbf{A}^\star$, the predictor is intentionally conventional; other GNN backbones can be substituted without altering the framework.

Table 1: AUROC (%) comparison on 10 benchmark datasets. Best result per column is in **bold**, second best is underlined.

| Category | Methods | Reddit | Weibo | Amaz. | Yelp. | T-Fin. | Ellip. | Tolo. | Quest. | DGra. | T-Soc. | Ave. |
|---|---|---|---|---|---|---|---|---|---|---|---|---|
| Classical ML | MLP | 60.71 | 92.89 | 96.94 | 81.64 | 92.24 | 88.32 | 73.78 | 60.15 | 72.34 | 73.06 | 79.21 |
| | KNN | 62.53 | 93.57 | 94.91 | 84.60 | 92.69 | 88.03 | 72.09 | 64.00 | 58.50 | 77.64 | 78.86 |
| | SVM | 60.79 | 98.77 | 97.53 | 93.80 | 94.85 | 91.68 | 74.14 | 54.85 | 70.37 | 79.94 | 81.67 |
| | XGBoost | 64.59 | 98.82 | 98.34 | 95.38 | 95.42 | 89.91 | 74.84 | 62.09 | 72.43 | 81.75 | 83.36 |
| | NA | 63.86 | 99.03 | 98.23 | 94.70 | 94.85 | 91.27 | 74.87 | 66.20 | 71.73 | 81.95 | 83.67 |
| Standard GNNs | GCN | 62.04 | 98.84 | 85.17 | 58.62 | 94.62 | 81.69 | 73.80 | 68.20 | 75.51 | 96.63 | 79.51 |
| | SGC | 66.95 | 98.58 | 87.84 | 57.66 | 87.97 | 79.52 | 72.36 | 68.11 | 68.63 | 83.67 | 77.13 |
| | GIN | 61.82 | 98.65 | 95.63 | 73.77 | 92.71 | 83.01 | 74.57 | 68.08 | 74.16 | 94.09 | 81.65 |
| | GraphSAGE | 63.28 | 97.75 | 90.90 | 82.90 | 95.62 | 87.59 | 80.92 | 73.04 | 75.60 | 95.73 | 84.33 |
| | GAT | 64.15 | 98.54 | 97.13 | 79.14 | 95.75 | 86.28 | 78.75 | 68.30 | 75.53 | 90.33 | 83.39 |
| | GT | 67.39 | 97.23 | 93.37 | 80.32 | 94.39 | 86.10 | 78.94 | 69.62 | 75.71 | 90.79 | 83.39 |
| | PNA | 63.45 | 98.78 | 82.14 | 74.39 | 92.47 | 84.86 | 75.17 | 66.32 | 73.39 | 84.02 | 79.50 |
| | BGNN | 68.94 | 98.59 | 90.64 | 63.28 | 96.20 | 87.69 | 76.65 | 68.02 | 76.22 | 99.94 | 82.62 |
| Specialised GNNs | GAS | 60.57 | 99.11 | 92.69 | 76.43 | 96.45 | 86.66 | 78.19 | 68.15 | 75.98 | 94.96 | 82.92 |
| | DCI | 66.87 | 97.94 | 95.39 | 78.38 | 87.85 | 85.73 | 73.86 | 64.77 | 73.94 | 83.96 | 80.87 |
| | PCGNN | 65.41 | 95.14 | 98.01 | 80.82 | 94.03 | 86.50 | 76.63 | 67.66 | 72.76 | 96.91 | 83.39 |
| | BernNet | 69.83 | 98.00 | 95.79 | 83.54 | 96.67 | 87.54 | 77.51 | 68.58 | 73.58 | 93.73 | 84.48 |
| | AMNet | 69.60 | 98.32 | 96.92 | 81.90 | 96.38 | 83.66 | 75.16 | 66.05 | 72.99 | 92.50 | 83.35 |
| | BWGNN | 70.82 | 98.13 | 98.27 | 87.13 | 96.93 | 87.03 | 80.41 | 70.87 | 76.30 | 96.88 | 86.28 |
| | GHRN | 61.02 | 99.18 | 98.29 | 84.60 | 96.46 | 89.50 | 80.08 | 72.16 | 76.13 | 97.12 | 85.45 |
| | PMP | 58.85 | 96.57 | 97.57 | 89.13 | 97.02 | 80.24 | 76.15 | 51.64 | 75.55 | 98.55 | 82.13 |
| | ConsisGAD | 62.58 | 94.72 | 95.51 | 76.16 | 95.02 | 83.48 | 73.32 | 70.38 | 67.10 | 94.96 | 81.32 |
| | GGAD | 62.29 | 79.23 | 94.43 | 53.08 | 82.28 | 74.90 | 44.83 | 42.23 | 47.92 | 85.71 | 66.69 |
| Combined Methods | RFGraph | 65.35 | 99.43 | 96.73 | 95.24 | 97.28 | 93.21 | 81.88 | 64.77 | 67.78 | 99.69 | 86.14 |
| | XGBGraph | 64.74 | 99.29 | 98.74 | 97.37 | 97.15 | 91.78 | 82.85 | 71.02 | 75.83 | 99.76 | 87.85 |
| | DGA-GNN | 59.79 | 99.22 | 97.54 | 97.76 | 97.66 | 93.62 | 82.43 | 71.82 | 76.08 | 99.91 | 87.58 |
| | GAAP | 68.73 | 99.05 | 97.76 | 98.55 | 97.47 | 92.05 | 84.74 | 71.24 | 76.57 | 99.96 | 88.61 |
| **Proposed** | **MAPGA** | **72.79** | **99.54** | **98.83** | **98.99** | **97.90** | **99.20** | **85.75** | **73.37** | **80.01** | **99.96** | **90.63** |

## 4 EXPERIMENTS

### 4.1 EXPERIMENTAL SETUP

**Datasets.** Experiments are carried out on **10** publicly available graph–anomaly benchmarks. **Reddit** (Kumar et al., 2019; Liu et al., 2022), **Weibo** (Kumar et al., 2019), **Amazon** (McAuley & Leskovec, 2013), **YelpChi** (Rayana & Akoglu, 2015), **Tolokers** (Platonov et al., 2023), **T-Finance** and **T-Social** (Tang et al., 2022), **Elliptic** (Weber et al., 2019), **DGraph-Fin** (Huang et al., 2022), and **Questions** (Platonov et al., 2023). Dataset details are provided in Appendix A.1.

**Baselines.** To ensure a comprehensive and unambiguous comparison, baselines are *not* grouped by the conceptual axes of *node feature embedding* and *global dependency modeling*, because most existing methods integrate both components, making such a split ambiguous. Instead, baselines are organized by architectural structure into four categories: *(1) Classical machine-learning (ML) models*: multilayer perceptron (MLP) (Rosenblatt, 1958), k-nearest neighbors (KNN) (Cover & Hart, 1967), support vector machine (SVM) (Chang & Lin, 2011), XGBoost (Chen & Guestrin, 2016), and Neighborhood Aggregation (NA) (Yang et al., 2023). *(2) Standard graph neural networks (GNNs)*: GCN (Kipf & Welling, 2016), SGC (Wu et al., 2019), GIN (Xu et al., 2018), GraphSAGE (Hamilton et al., 2017), GAT (Veličković et al., 2017), GT (Shi et al., 2020), PNA (Corso et al., 2020), and BGNN (Ivanov & Prokhorenkova, 2021). *(3) Anomaly-oriented GNNs*: GAS (Li et al., 2019), DCI (Wang et al., 2021), PC-GNN (Liu et al., 2021), BernNet (He et al., 2021), AMNet (Chai et al., 2022), BWGNN (Tang et al., 2022), GHRN (Gao et al., 2023), ConsisGAD (Chen et al., 2024b), GGAD (Qiao et al., 2024) and PMP (Zhuo et al., 2024). *(4) Command Methods* that combine ML with GNN architectures: RFGraph and XGBGraph from GAD-Bench (Tang et al., 2023), Dynamic Grouping Aggregation GNN (DGA-GNN) (Duan et al., 2024), and GAAPA (Duan et al., 2025).

Table 2: The results (AUROC, %) of the ablation study on different modules. The highest score is highlighted in **bold**.

| Model | Reddit | Weibo | Amaz. | Yelp. | T-Fin. | Ellip. | Tolo. | Quest. | DGra. | T-Soc. | Ave. |
|---|---|---|---|---|---|---|---|---|---|---|---|
| w/o MAPE | 65.48 | 92.89 | 96.94 | 81.64 | 92.24 | 88.32 | 73.78 | 60.15 | 72.34 | 73.06 | 79.58 |
| w XGBoost | 63.67 | 93.78 | 89.82 | 50.00 | 96.45 | 87.82 | 64.51 | 54.44 | 72.34 | 56.77 | 72.97 |
| w/o MPGA | 67.09 | 98.99 | 95.93 | 91.95 | 95.89 | 95.10 | 82.90 | 68.26 | 74.87 | 97.38 | 86.83 |
| w GCN | 70.88 | 98.42 | 96.26 | 96.46 | 96.18 | 97.80 | 83.14 | 71.40 | 77.24 | 99.93 | 88.77 |
| w GAT | 66.49 | 98.65 | 97.31 | 95.41 | 97.42 | 98.78 | 83.50 | 72.50 | 71.08 | 91.23 | 87.24 |
| MDPGA | **72.79** | **99.54** | **98.83** | **98.99** | **97.90** | **99.20** | **85.75** | **73.37** | **80.01** | **99.96** | **90.63** |

**Evaluation Metrics.** Three metrics are considered to assess the performance of MDPE: AUROC, AUPRC, and Rec@$K$. **AUROC** (area under the ROC) quantifies the ability to distinguish positive from negative classes and is widely regarded as a stable, threshold-agnostic indicator of discrimination performance. **AUPRC** (area under the PRC) summarizes the trade-off between precision and recall across thresholds and is particularly informative under severe class imbalance. **Rec@$K$** (recall at rank $K$) measures the proportion of true anomalies retrieved among the top-$K$ ranked instances, where $K$ equals the number of anomalous samples in the test set. Owing to space constraints, AUROC is adopted as the primary metric and its results are reported in the main text; complete AUPRC and Rec@$K$ outcomes are deferred to the Appendix for reference.

The *source codes* and more results about *AUPRC*, *Rec@K* are given in the *supplementary materials*.

## 4.2 COMPARISON WITH SOTA

As summarized in Table 1, MAPGA was evaluated on 10 public benchmarks against 27 state-of-the-art baselines spanning classical ML, standard GNNs, specialized GNNs, and combined methods, and achieved the best (or tied best) AUROC on all datasets, with the highest average AUROC of 90.63% (+2.02 percentage points (pp) over the strongest prior, GAP at 88.61%). On Elliptic, MAPGA reached 99.20% AUROC, exceeding the best baseline (DGA-GNN, 93.62%) by +5.58pp and approaching a perfect score; it also delivered near-perfect results on Weibo (99.54%), Elliptic (99.20%), and T-Social (99.96%), demonstrating robust and consistent superiority across diverse graph anomaly detection scenarios.

## 4.3 ABLATION STUDY

The ablation study investigates the contributions of two core components (MAPE and MPGA) and the effects of two hyperparameters: view count and embedding dimension, while deviation and computational complexity are discussed in Appendix A.5 and Appendix A.6.

**Effectiveness of MAPE and MPGA.** The ablation results in Table 2 corroborate the indispensable roles of both modules. Eliminating the **MAPE** encoder (*w/o MAPE*) lowers the average AUROC from 90.63% to 79.58% (-11.05p), and substituting it with an XGBoost-based leaf embedding (*w XGBoost*) causes an even steeper decline of 17.66 pp. These pronounced drops highlight the superiority of the proposed encoder for node representation. Conversely, removing the **MPGA** propagator (*w/o MPGA*) results in a modest 3.8pp reduction, indicating that MAPE alone already supplies strong discriminative signals; however, replacing MPGA with either a vanilla GCN (*w GCN*) or a vanilla GAT (*w GAT*) degrades the mean AUROC by 1.86pp and 3.39pp, respectively, demonstrating the importance of a tailored propagation strategy. The complete model (denoted as *MAPGA* in the table) secures the best performance on every dataset. Collectively, these observations confirm that MAPE delivers the primary representational gains, while MPGA provides complementary propagation benefits essential for attaining state-of-the-art anomaly detection accuracy.

**Impact of the Number of Views $K$.** The **top row** of Figure 4 illustrates how the two evaluation metrics—AUPRC (solid blue) and Rec@$K$ (dashed orange)—vary as the number of views $K$ is increased from 10 to 70 on four representative benchmarks. A consistent upward trend is observed on

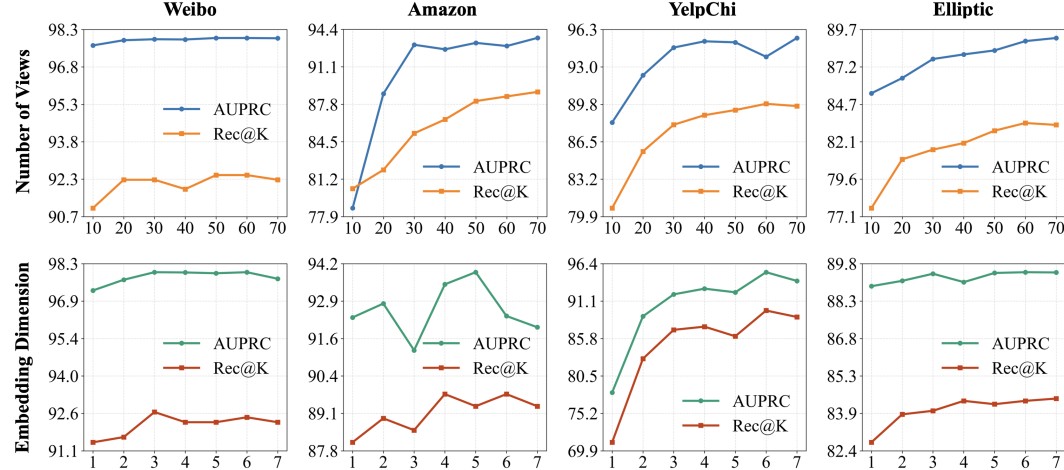

Figure 4: **Effect of view count and embedding size.** Detection performance (AUPRC and Rec@$K$) versus the number of views $K$ (top row) and the per-view embedding dimension $d_0$ (bottom row) on four representative benchmarks.

every dataset: accuracy rises sharply between $K{=}10$ and $K \approx 40$, confirming that each additional view injects complementary partitions that enrich the discrete semantics captured by MDP-E. Beyond $K \approx 40$ the curves flatten, indicating diminishing returns once the feature manifold has been sufficiently covered. Consequently, a moderate choice of $K{=}40$–$50$ is adopted in all subsequent experiments to balance performance and computational overhead.

**Impact of Per-view Embedding Dimension $d_0$.** **Bottom row** of Fig. 4 evaluates the influence of the per-view embedding dimension $d_0$ by sweeping $d_0 \in \{1, \ldots, 7\}$. Both AUPRC (green) and Rec@$K$ (red) increase markedly as $d_0$ grows from 1 to 4, after which the curves plateau or fluctuate slightly. These results indicate that $d_0 = 4$ supplies sufficient representational capacity for the discrete sub-spaces, whereas larger values yield negligible gains while roughly doubling the parameter count. Consequently, $d_0$ is fixed to 4 in all subsequent experiments.

## 4.4 DISSCUSSION.

Across the ten benchmark datasets, the proposed framework exceeds every state-of-the-art baseline, posting a mean AUROC that is **+2.02,pp** above the strongest competitor (Table 1). The ablation study shows that each module is indispensable: removing the Multi-view Discrete Partition Encoder (*w/o MDP-E*) reduces mean AUROC by **11.05,pp** (Table 2), confirming that learnable multi-view discrete partitioning is crucial for restoring manifold connectivity. Re-introducing the Behaviour-Pattern Global Association adds a further **+3.8,pp** (with a peak of **+6.3,pp** on *Reddit*), indicating that global behaviour-pattern cues complement the local gains from MDP-E by unveiling residual anomalies. Collectively, these findings substantiate the synergistic effect of dynamic partitioning and behaviour-pattern association in advancing graph-based anomaly detection.

## 5 CONCLUSION

This work introduces MAPGA, a novel GAD framework that *simultaneously* restores manifold connectedness and discovers long-range dependencies. MAPGA first uses a learnable multi view partition encoder to convert fragmented features into coherent discrete embeddings. It then fuses graphs built from common subspace co-occurrence and neighbourhood behaviour histograms with the original topology, enriching the context for GCN inference. Extensive experiments on public benchmarks confirm that MAPGA outperforms state-of-the-art methods. The main limitation lies in the computational overhead introduced by multiple parallel adaptive partition operators. Future work will merge redundant subspaces and compress the model to support large scale application.

## ETHICS STATEMENT

This work adheres to the ICLR Code of Ethics. Experiments rely on publicly available or appropriately licensed datasets; where data may contain personal or sensitive attributes, de-identification and license terms are respected, and no attempt is made to re-identify individuals. The study does not target or enable discriminatory or unsafe use; foreseeable dual-use risks are discussed and mitigation strategies (e.g., responsible release, robustness and bias checks) are described in the supplementary materials. No human-subject intervention, clinical decision-making, or deployment in safety-critical settings was conducted; any future deployment will follow applicable legal and institutional review requirements. Funding sources and potential conflicts of interest are disclosed. All procedures, data handling, and reporting were conducted with attention to privacy, fairness, and research integrity.

## REPRODUCIBILITY STATEMENT

Reproducibility has been prioritized. The main text specifies datasets and splits, preprocessing pipelines, model architectures, training schedules, and evaluation protocols, while exact hyperparameters, random seeds, and ablation configurations are provided in the appendix. An anonymized repository with source code, configuration files, and scripts to regenerate all tables and figures is supplied in the supplementary materials, including environment setup instructions and hardware requirements. For theoretical results, assumptions are stated in the paper and complete proofs are included in the appendix. External baselines are referenced with versions or commit hashes, and dataset licenses and checksums are reported to ensure fidelity. Together, these materials enable end-to-end reproduction of the reported results.

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

## A APPENDIX

The supplementary material provides additional information, including comprehensive datasets descriptions (*Section A.1*), decision boundary visualizations (*Section A.2*), further comparative experiments (*Section A.3*) and disclosure of language model assistance (*Section A.4*). The source code for the proposed method is provided in the `code` file.

## A.1 COMPREHENSIVE DATASETS DESCRIPTION

This section provides detailed information on the ten public datasets employed in this study, and Table 3 summarizes their key characteristics. For all datasets, the train–test split follows the official ratios and is generated by random sampling; the exact splitting script is provided in our released code and referenced in the supplementary material.

Table 3: Overview of datasets with their corresponding node feature types, feature dimension, and detailed descriptions

| Dataset | #Nodes | #Edges | #Feat. | Anomaly | Train | Relation Concept | Feature Type |
|---------|--------|--------|--------|---------|-------|------------------|--------------|
| Reddit | 10,984 | 168,016 | 64 | 3.3% | 40% | Under Same Post | Text Embedding |
| Weibo | 8,405 | 407,963 | 400 | 10.3% | 40% | Under Same Hashtag | Text Embedding |
| Amazon | 11,944 | 4,398,392 | 25 | 9.5% | 70% | Review Correlation | Misc. Information |
| YelpChi | 45,954 | 3,846,979 | 32 | 14.5% | 70% | Reviewer Interaction | Misc. Information |
| Tolokers | 11,758 | 519,000 | 10 | 21.8% | 40% | Work Collaboration | Misc. Information |
| Questions | 48,921 | 153,540 | 301 | 3.0% | 52% | Question Answering | Text Embedding |
| T-Finance | 39,357 | 21,222,543 | 10 | 4.6% | 50% | Transaction Record | Misc. Information |
| Elliptic | 203,769 | 234,355 | 166 | 9.8% | 50% | Payment Flow | Misc. Information |
| DGraph-Fin | 3,700,550 | 4,300,999 | 17 | 1.3% | 70% | Loan Guarantor | Misc. Information |
| T-Social | 5,781,065 | 73,105,508 | 10 | 3.0% | 40% | Social Friendship | Misc. Information |

**Reddit**. This dataset presents a user–subreddit interaction graph, capturing posts shared across various subreddits over the course of one month. It includes verified labels for banned users. Focusing on the 1,000 most active subreddits and the 10,000 most engaged users, the dataset contains a total of 672,447 interactions. Posts are transformed into feature vectors using Linguistic Inquiry and Word Count (LIWC) categories.

**Weibo**. This dataset is constructed from the Tencent Weibo platform, consisting of a graph linking 8,405 users with 61,964 hashtags. Suspicious activity is defined as posting twice within specific short time intervals (e.g., 60 seconds). Users who engaged in five or more such incidents are labeled as "suspicious"; all others are labeled "benign". In total, the dataset includes 868 suspicious and 7,537 benign users. Feature vectors incorporate location data and bag-of-words representations of the posts.

**YelpChi**. Designed for the detection of deceptive reviews on Yelp.com, this dataset includes a graph with three types of edges: R-U-R (reviews posted by the same user), R-S-R (reviews for the same product with the same star rating), and R-T-R (reviews for the same product posted in the same month). These connections help identify reviews that may unfairly promote or defame a product or business.

**Amazon**. The Amazon dataset targets the detection of users paid to write fake reviews in the Musical Instruments category on Amazon.com. The graph encodes three types of relationships: U-P-U (users reviewing at least one common product), U-S-U (users giving at least one same star rating within one week), and U-V-U (users with top-5% mutual review similarity).

**T-Finance**. Focused on detecting anomalous accounts in transaction networks, this dataset contains nodes representing anonymized user accounts. Each node has 10-dimensional features capturing registration duration, login activity, and interaction frequency. Edges represent transactional relationships. Human experts labeled accounts involved in fraudulent or illicit activity, including money laundering and online gambling.

**Tolokers**. Collected from the Toloka crowdsourcing platform, this dataset includes nodes representing workers involved in at least one of 13 selected projects. Edges connect workers who collaborated on the same task. The task is to predict which workers were banned from any of the projects. Node features are derived from user profiles and performance statistics.

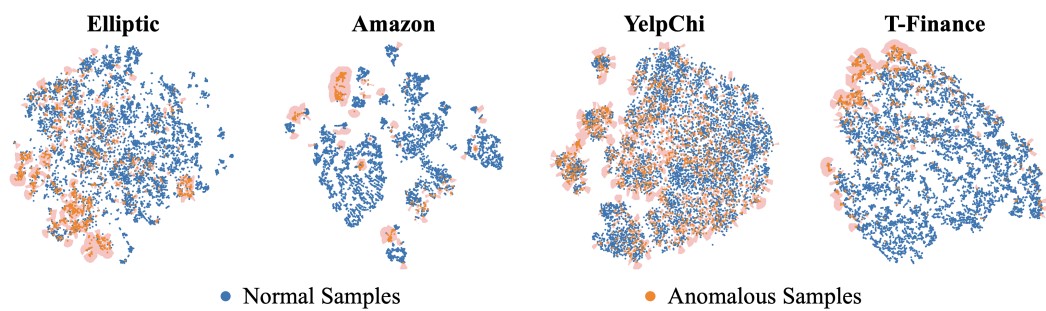

| Elliptic | Amazon | YelpChi | T-Finance |

● Normal Samples        ● Anomalous Samples

Figure 5: Decision-boundary visualisation of the proposed MAPGA model on four datasets, shown as t-SNE projections of the learned embeddings. Blue points represent normal nodes and orange points the anomalies detected by MAPGA. Despite the markedly fragmented manifolds—where samples of the same class break into disconnected sub-regions—the model maintains locally smooth decision surfaces within each fragment and preserves clear inter-fragment separation, thereby uncovering sparse, latent anomaly clusters that conventional methods fail to expose.

**Questions**. This dataset comes from Yandex Q, a question-answering platform. Nodes represent users, with edges connecting users if one answered a question posed by the other between September 2021 and August 2022. The focus is on users interested in the topic "medicine", and the goal is to predict which users remained active by the end of the collection period. Features include the average FastText embedding of words in user descriptions, along with a binary indicator for users without a description.

**Elliptic**. This dataset contains a graph of over 200,000 Bitcoin transactions (nodes) and 234,000 directed payment flows (edges), along with 166 features per node. It maps transactions to real-world entities, categorizing them as licit (e.g., exchanges, miners, wallet providers) or illicit (e.g., scams, ransomware, Ponzi schemes, terrorist organizations).

**T-Social**. Aimed at identifying anomalous accounts in social networks, this dataset shares node features and annotations with T-Finance. An edge connects two accounts if they maintained a friendship for more than three months. Anomalous nodes were labeled by human experts and are associated with behaviors such as fraud, money laundering, and online gambling.

**DGraph-Fin**. Provided by Finvolution Group, this large-scale dynamic social graph represents users in the financial sector. Nodes are Finvolution users, and edges indicate emergency contact relationships. The objective is to identify anomalous users—specifically, those with overdue behavior. The dataset contains over 3 million nodes, 4 million dynamic edges, and more than 1 million ground-truth labels, making it highly imbalanced.

## A.2 Decision Boundary Visualizations

Figure 5 presents t-SNE projections of the learned embeddings on four representative datasets, where blue points denote normal nodes and orange points indicate the anomalies detected by MAPGA. All benchmarks exhibit *manifold fragmentation*: samples from the same class are torn into multiple, disconnected sub-manifolds, thereby violating the local-smoothness assumption underpinning conventional deep models. Even under this adverse topology, MAPGA succeeds in tracing coherent decision boundaries: (i) within each isolated sub-manifold the margin remains locally smooth and tightly encloses latent anomaly pockets, while (ii) across sub-manifolds it preserves clear inter-class separation without resorting to over-bending the hypersurface. These behaviours stem from the synergy of MAPGA's *multi-view adaptive partitioning*, which decomposes the global space into fine-grained, locally Euclidean cells, and its *cross-view global association* module, which aligns the embeddings of disparate cells to compensate for information gaps created by the fractures. The ac-

companying multi-scale regularisation further enlarges confidence margins in low-density regions, enabling the framework to expose subtle, sparsely distributed anomaly patterns. Collectively, the evidence demonstrates MAPGA's strong capacity to model highly fragmented manifolds and to excavate hidden abnormal structures even when manifold connectivity is fundamentally absent.

## A.3 ADDITIONAL COMPARATIVE EXPERIMENTS

To complement the AUROC comparison, we conduct additional large-scale evaluations using AUPRC and Rec@$K$, two metrics that emphasize positive-class precision and top-$K$ retrieval under severe class imbalance. Following the same protocol as the AUROC study, MAPGA is compared against 27 competitive baselines spanning classical ML, standard GNNs, specialized GNNs, and combined methods across 10 benchmarks. As reported in Tables 4 and 5, MAPGA attains the best average AUPRC (66.33%, +1.93pp over GAP) and the best average Rec@K (65.01%, +2.98pp over GAP), providing a complementary view of its superiority beyond threshold-independent AUROC.

Table 4: AUPRC (%) comparison on 10 benchmark datasets. Best result per column is in **bold**, second best is underlined.

| Category | Model | Reddit | Weibo | Amaz. | Yelp. | T–Fin. | Ellip. | Tolo. | Quest. | DGra. | T–Soc. | Ave. |
|---|---|---|---|---|---|---|---|---|---|---|---|---|
| Classical | MLP | 5.91 | 84.88 | 87.34 | 47.68 | 74.21 | 43.77 | 38.29 | 15.34 | 2.69 | 9.69 | 40.98 |
| | KNN | 6.12 | 81.12 | 84.41 | 54.39 | 74.97 | 60.98 | 35.30 | 15.37 | 1.67 | 36.32 | 45.07 |
| | SVM | 6.88 | 84.91 | 85.80 | 41.01 | 78.10 | 20.98 | 37.90 | 15.37 | 2.65 | OOT | 41.51 |
| | XGBoost | 5.56 | 94.49 | 91.88 | 84.00 | 82.64 | 76.93 | 40.05 | 16.24 | 2.75 | 16.60 | 51.11 |
| | NA | 9.70 | 94.09 | 91.56 | 63.93 | 88.78 | 29.14 | 51.06 | 14.32 | 4.13 | 79.21 | 52.59 |
| Standard GNNs | GCN | 4.63 | 94.64 | 45.65 | 20.88 | 78.22 | 25.37 | 40.57 | 14.06 | 3.80 | 76.35 | 40.42 |
| | SGC | 6.04 | 91.16 | 42.69 | 19.87 | 68.68 | 17.82 | 39.59 | 10.53 | 2.49 | 16.28 | 31.52 |
| | GIN | 6.41 | 91.67 | 84.61 | 33.63 | 78.35 | 26.21 | 40.36 | 13.68 | 3.47 | 60.79 | 43.92 |
| | GraphSAGE | 5.56 | 94.02 | 82.45 | 46.64 | 84.71 | 57.82 | 51.41 | 17.50 | 3.77 | 75.32 | 51.92 |
| | GAT | 7.20 | 92.91 | 87.94 | 43.62 | 82.72 | 27.53 | 45.25 | 15.51 | 3.85 | 32.07 | 43.86 |
| | GT | 7.68 | 89.85 | 84.90 | 44.60 | 83.14 | 25.90 | 45.71 | 17.08 | 3.83 | 36.14 | 43.88 |
| | PNA | 7.75 | 96.04 | 35.24 | 29.95 | 76.67 | 27.81 | 41.74 | 11.38 | 3.22 | 21.24 | 35.10 |
| | BGNN | 6.87 | 95.99 | 67.92 | 29.71 | 83.72 | 62.03 | 45.35 | 9.43 | 4.24 | 99.09 | 50.44 |
| Specialised GNNs | GAS | 4.43 | 96.76 | 81.43 | 35.11 | 85.95 | 29.80 | 47.21 | 15.48 | 3.65 | 62.36 | 46.22 |
| | DCI | 7.74 | 91.77 | 85.17 | 39.88 | 63.68 | 27.39 | 37.73 | 14.59 | 3.31 | 12.97 | 38.42 |
| | PCGNN | 7.73 | 89.07 | 89.33 | 44.51 | 83.31 | 42.66 | 44.85 | 15.59 | 3.42 | 80.29 | 50.08 |
| | BernNet | 7.82 | 92.38 | 84.89 | 51.92 | 89.17 | 38.25 | 43.69 | 17.25 | 3.27 | 44.30 | 47.29 |
| | AMNet | 7.87 | 94.99 | 88.36 | 46.86 | 88.87 | 25.18 | 40.74 | 15.63 | 2.81 | 37.70 | 44.90 |
| | BWGNN | 8.32 | 94.01 | 91.48 | 61.53 | 89.38 | 29.31 | 49.58 | 18.57 | 3.97 | 78.93 | 52.51 |
| | GHRN | 4.66 | 95.27 | 89.52 | 55.42 | 87.60 | 43.90 | 47.45 | 18.31 | 3.80 | 86.78 | 53.27 |
| | PMP | 4.26 | 89.54 | 89.29 | 68.94 | 90.26 | 40.12 | 43.34 | 4.01 | 2.63 | 87.68 | 52.01 |
| | ConsisGAD | 5.69 | 82.95 | 83.86 | 36.66 | 80.78 | 15.61 | 37.79 | 15.38 | 0.65 | 46.48 | 40.54 |
| | GGAD | 5.06 | 55.76 | 62.27 | 17.40 | 2.77 | 13.21 | 19.24 | 2.46 | 1.19 | 20.57 | 19.99 |
| Combined Methods | RFGraph | 5.13 | 96.95 | 90.53 | 83.92 | 89.23 | 78.86 | 52.34 | 14.44 | 2.15 | 97.63 | 61.12 |
| | XGBGraph | 5.29 | 97.06 | 93.33 | 91.11 | 90.12 | 77.78 | 53.92 | 18.19 | 3.79 | 97.34 | 62.79 |
| | DGA–GNN | 6.82 | 97.14 | 88.95 | 91.31 | 90.00 | 77.71 | 54.12 | 18.66 | 3.90 | 98.99 | 62.76 |
| | GAP | 7.53 | 97.16 | 92.58 | 94.96 | 90.79 | 79.59 | 58.39 | **19.43** | 4.17 | 99.42 | 64.40 |
| **Proposed** | **MAPGA** | **11.15** | **97.98** | **93.87** | **95.68** | **91.43** | **89.93** | **60.25** | 19.11 | **4.33** | **99.58** | **66.33** |

In Table 4, MAPGA delivers the highest average AUPRC of 66.33%, surpassing the strongest prior (GAP, 64.40%) by +1.93 percentage points (pp). Column-wise, MAPGA attains the best result on 9/10 datasets and ranks second on *Quest* (19.11% vs. 19.43% for GAP, $-0.32$pp). Notably, it achieves near-perfect performance on *Weibo* (97.98%) and *T-Social* (99.58%), and yields a large gain on *Elliptic* (89.93% vs. 79.59% for GAP, +10.34pp). Consistent improvements are also observed on *Amazon* (+0.54pp), *Yelp* (+0.72pp), *T-Fin.* (+0.64pp), *Tolo.* (+1.86pp), *Reddit* (+1.45pp), and *DGra.* (+0.09pp), indicating robust advantages across diverse anomaly detection regimes.

In Table 5, MAPGA attains the best average Rec@K of 65.01%, outperforming the strongest baseline (GAP, 62.03%) by +2.98pp. It secures the top score on 7/10 datasets (*Amazon*, *Yelp*, *T-Fin.*, *Elliptic*, *Tolo.*, *Quest.*, *T-Social*) and is second on *Reddit* (13.73% vs. 14.97% for PCGNN). Performance on *Weibo* remains competitive (92.62% vs. 93.37% for BGNN), while *DGra.* is below the peak baseline (7.03% vs. 7.73% for GAP). Importantly, MAPGA delivers substantial gains on *Ellip-

Table 5: Rec@K (%) comparison on 10 benchmark datasets. Best result per column is in **bold**, second best is underlined.

| Category | Model | Reddit | Weibo | Amaz. | Yelp. | T–Fin. | Ellip. | Tolo. | Quest. | DGra. | T–Soc. | Ave. 10 |
|---|---|---|---|---|---|---|---|---|---|---|---|---|
| Classical | MLP | 9.52 | 78.39 | 83.15 | 46.62 | 68.93 | 57.43 | 39.88 | 16.99 | 4.04 | 16.86 | 42.18 |
| | KNN | 10.20 | 73.20 | 79.35 | 51.92 | 70.04 | 56.60 | 37.85 | 19.18 | 1.98 | 44.08 | 44.44 |
| | SVM | 5.44 | 84.15 | 86.41 | 70.23 | 75.59 | 72.36 | 39.72 | 17.81 | 4.21 | 45.48 | 50.18 |
| | XGBoost | 6.80 | 87.03 | 86.41 | 75.08 | 76.01 | 72.58 | 41.90 | 16.71 | 4.30 | 21.86 | 48.87 |
| | NA | 8.84 | 85.59 | 87.50 | 73.38 | 75.31 | 71.74 | 42.21 | 18.36 | 4.04 | 21.70 | 48.87 |
| Standard GNN | GCN | 6.12 | 88.47 | 44.02 | 23.85 | 74.90 | 33.52 | 39.41 | 17.81 | 7.05 | 73.23 | 40.84 |
| | SGC | 8.84 | 87.32 | 46.20 | 22.46 | 67.41 | 22.99 | 39.72 | 16.99 | 3.87 | 24.93 | 34.07 |
| | GIN | 10.88 | 89.63 | 80.98 | 36.15 | 73.37 | 32.13 | 39.41 | 18.36 | 6.32 | 64.47 | 45.17 |
| | GraphSAGE | 7.48 | 90.78 | 76.26 | 27.15 | 78.09 | 56.69 | 48.75 | 21.37 | 6.84 | 73.74 | 50.92 |
| | GAT | 10.88 | 89.63 | 82.61 | 44.23 | 79.75 | 37.86 | 44.24 | 17.26 | 7.14 | 72.07 | 45.57 |
| | GT | 12.93 | 85.88 | 78.80 | 44.62 | 81.55 | 30.75 | 44.39 | 20.27 | 6.92 | 43.58 | 44.97 |
| | PNA | 10.88 | 39.67 | 90.78 | 30.00 | 72.54 | 36.47 | 42.68 | 13.15 | 5.16 | 26.95 | 36.83 |
| | BGNN | 9.52 | 93.37 | 64.67 | 30.77 | 77.25 | 59.56 | 45.33 | 12.60 | 7.70 | 96.89 | 49.77 |
| Specialised GNN | GAS | 4.08 | 91.93 | 80.43 | 38.00 | 79.75 | 37.49 | 47.04 | 18.90 | 6.02 | 64.58 | 46.82 |
| | DCI | 8.16 | 89.05 | 80.98 | 40.46 | 71.98 | 35.27 | 37.85 | 17.26 | 5.85 | 18.27 | 40.51 |
| | PCGNN | **14.97** | 84.15 | 85.33 | 43.77 | 79.06 | 43.15 | 43.77 | 19.73 | 6.66 | 73.53 | 49.25 |
| | BernNet | 10.88 | 89.34 | 82.61 | 49.69 | 83.63 | 43.77 | 44.69 | 19.73 | 4.21 | 43.21 | 48.41 |
| | AMNet | 11.56 | 88.76 | 83.13 | 45.38 | 84.05 | 30.56 | 42.52 | 17.53 | 4.21 | 43.21 | 45.09 |
| | BWGNN | 11.56 | 87.90 | 85.87 | 56.69 | 84.19 | 42.47 | 50.31 | 21.64 | 7.57 | 75.78 | 52.40 |
| | GHRN | 5.44 | 89.34 | 85.33 | 51.85 | 81.97 | 50.51 | 46.57 | 22.19 | 6.96 | 82.33 | 52.25 |
| | PMP | 2.04 | 85.30 | 83.15 | 61.69 | 85.99 | 46.64 | 44.39 | 5.75 | 3.31 | 81.11 | 49.94 |
| | ConsisGAD | 9.16 | 86.31 | 88.34 | 64.52 | 87.47 | 49.44 | 60.41 | 5.86 | 4.99 | 77.02 | 53.35 |
| | GGAD | 7.73 | 20.77 | 54.17 | 14.38 | 2.68 | 25.37 | 5.60 | 5.03 | 2.55 | 42.31 | 18.06 |
| Combined Methods | RFGraph | 3.40 | 91.93 | 83.15 | 75.31 | 84.05 | 72.58 | 52.18 | 15.89 | 3.22 | 93.58 | 57.53 |
| | XGBGraph | 6.12 | 91.93 | 85.87 | 83.15 | 85.02 | 71.93 | 53.43 | 20.55 | 6.96 | 95.33 | 59.85 |
| | DGA–GNN | 9.52 | 90.78 | 85.87 | 84.23 | 84.33 | 73.72 | 55.14 | 19.73 | 7.52 | 95.97 | 60.59 |
| | GAP | 10.88 | 92.22 | 87.50 | 88.54 | 85.71 | 73.32 | 56.08 | 21.10 | **7.73** | 97.25 | 62.03 |
| **Proposed** | **MAPGA** | 13.73 | **92.62** | **89.75** | **90.20** | **88.67** | **85.16** | **60.67** | **24.73** | 7.03 | **97.53** | **65.01** |

*tic* (85.16% vs. 73.72% for *DGA-GNN*, +11.44pp) and near-perfect retrieval on *T-Social* (97.53%), underscoring consistent superiority on challenging and highly imbalanced graphs.

Across both metrics, MAPGA wins on the majority of datasets, achieves near-ceiling results on *Weibo* and *T-Social* (AUPRC 97.98% and 99.58%; Rec@K 97.53%), and delivers especially strong gains on *Elliptic* (+10.34pp AUPRC and +11.44pp Rec@K over the strongest baseline). These consistent improvements across diverse graph types and baseline families, together with the AUROC results in Table 1, offer convergent evidence that MAPGA is both robust and reliable for graph anomaly detection in practice.

## A.4    DISCLOSURE OF LANGUAGE MODEL ASSISTANCE

Large language models were used only for editorial polishing (grammar, style, and minor rephrasing). They were not used for research design, methods, analysis, coding, figures/tables, or references. All scientific content was authored and verified by the authors, and all edits were manually reviewed. This use does not meet contributorship thresholds and does not affect reproducibility.

## A.5    DEVIATION ANALYSIS OF MAPGA

To assess the robustness and stability of MAPGA under different random initializations, we report the deviation (mean $\pm$ standard deviation) across multiple runs on all datasets. Table A.4 summarizes the variance of AUROC, AUPRC, and Rec@K.

Overall, the results show that MAPGA exhibits consistently low variance on most datasets, indicating stable optimization behaviour. In high-density or large-scale graphs such as *Weibo*, *Amazon*, and *T-Finance*, the deviation remains below 0.5 for AUROC and AUPRC. Datasets with inherently higher heterophily or sparse anomaly supervision, such as *Tolokers* and *Questions*, show slightly

Table 6: Performance deviation (mean $\pm$ std) of MAPGA across all datasets.

| Metrics | Reddit | Weibo | Amazon | YelpChi | T-Finance | Elliptic | Tolokers | Questions | DGra. | T-Social |
|---|---|---|---|---|---|---|---|---|---|---|
| **AUROC** | $71.22 \pm 0.57$ | $99.51 \pm 0.03$ | $98.77 \pm 0.06$ | $98.76 \pm 0.23$ | $97.86 \pm 0.04$ | $99.15 \pm 0.04$ | $85.20 \pm 0.55$ | $73.15 \pm 0.22$ | $78.80 \pm 0.21$ | $99.96 \pm 0.01$ |
| **AUPRC** | $10.96 \pm 0.19$ | $97.89 \pm 0.09$ | $93.60 \pm 0.27$ | $95.25 \pm 0.43$ | $91.22 \pm 0.21$ | $88.84 \pm 1.09$ | $59.10 \pm 1.15$ | $18.06 \pm 1.05$ | $4.32 \pm 0.01$ | $99.46 \pm 0.12$ |
| **Rec@K** | $13.11 \pm 0.62$ | $92.13 \pm 0.49$ | $89.74 \pm 0.01$ | $88.80 \pm 0.40$ | $88.39 \pm 0.28$ | $84.36 \pm 0.80$ | $60.23 \pm 0.44$ | $23.72 \pm 1.01$ | $6.81 \pm 0.22$ | $87.26 \pm 0.27$ |

larger variance, which is expected due to their noisier attribute distributions. These observations confirm that MAPGA remains reliable across diverse graph types.

## A.6 COMPUTATIONAL COMPLEXITY

This section analyzes the computational complexity of MAPGA. We explicitly distinguish the *one-off initialisation* of the partition rules in MAPE from the *per-epoch* training cost of the full model. Importantly, the shallow decision trees described in Section **??** are used *only* to initialise the scalar split rules of MAPE; after initialisation, the trees are discarded and are neither evaluated nor updated during training or inference. The deployed model therefore consists of learnable scalar thresholds and embedding tables, rather than a tree–GNN ensemble.

### A.6.1 NOTATION

Let $G = (V, E)$ be the input graph with $n = |V|$ nodes and $m = |E|$ edges. Each node has an $N$-dimensional raw feature vector. MAPE instantiates $K$ views, and view $k$ contains $L_k$ partition rules; we denote the average number of rules by

$$\bar{L} = \frac{1}{K} \sum_{k=1}^{K} L_k.$$

Let $E_k \in \mathbb{R}^{2^{L_k} \times d_0}$ denote the codebook (embedding dictionary) for view $k$, where $d_0$ is the per-view embedding dimension. The final node embedding has dimension

$$d = \sum_{k=1}^{K} d_0 = K d_0.$$

We denote by $L$ the number of GCN layers used in the anomaly-detection backbone.

### A.6.2 OFFLINE INITIALISATION OF PARTITION RULES

For each view, MAPE defines $L_k$ scalar split rules of the form $(n_k^\ell, s_k^\ell)$, where $n_k^\ell$ selects one feature and $s_k^\ell$ is a threshold (see Section **??**). In our implementation, these thresholds are *initialised* from shallow decision trees: we train a tree $T_k$ on the raw feature space, and convert each internal split node into one scalar rule. After this conversion, the tree structure itself is discarded and only the thresholds $\{s_k^\ell\}$ remain as learnable parameters.

If a standard CART-style procedure is used, training one tree on $n$ samples with $N$ features costs $O(nN \log n)$ in the worst case. Initialising $K$ views in this way therefore costs

$$T_{\text{init}} = O(K\, n\, N \log n), \tag{9}$$

which is a one-time offline operation and does not appear in the per-epoch complexity. Alternative initialisation strategies (e.g., random thresholds) would leave the per-epoch complexity unchanged.

### A.6.3 MAPE ENCODER (PER EPOCH)

During each forward pass, view $k$ applies its $L_k$ scalar split rules to every node: each rule evaluates a threshold condition on a single feature. Generating all binary codes and integer indices in view $k$ therefore costs $O(nL_k)$, and across all views we obtain

$$T_{\text{MAPE-codes}} = O\left(n \sum_{k=1}^{K} L_k\right) = O(nK\bar{L}). \tag{10}$$

Given the integer index for node $i$ in view $k$, MAPE performs a table lookup in the view-specific codebook $E_k$, which is $O(1)$ per node and per view. These lookups are thus absorbed into the $O(nK\bar{L})$ term above. Hence, the total MAPE forward cost per epoch is

$$T_{\text{MAPE}} = O(nK\bar{L}). \tag{11}$$

No decision-tree evaluation is involved at this stage: MAPE uses only the learnable scalar rules and codebooks.

### A.6.4 MPGA Association Graphs (Per Epoch)

Given the per-view one-hot codes, MPGA constructs two auxiliary graphs: a representation-pattern graph $G_e$ and a behaviour-pattern graph $G_b$.

**Representation-pattern graph.** In each view, we measure whether two nodes share the same subspace assignment. A naive all-pairs computation would cost $O(Kn^2)$. In practice, we restrict attention to a sparsified set of candidate pairs (e.g., within local neighbourhoods or sampled pairs) and cap the number of non-zero affinities such that

$$|E_e| = O(m). \tag{12}$$

Under this sparsity assumption, the cost of constructing $G_e$ is linear in the number of retained affinities:

$$T_{\text{rep}} = O(K\,|E_e|) = O(Km). \tag{13}$$

**Behaviour-pattern graph.** For each node and each view, MPGA aggregates one-hot codes over its in- and out-neighbours to build histograms of neighbour assignments. Each edge contributes $O(K)$ additions (one per view), so the cost of computing all histograms is

$$T_{\text{beh-hist}} = O(Km). \tag{14}$$

Behavioural affinities (e.g., via cosine similarity) are then computed only for a sparsified set of node pairs, and the number of non-zero entries in the behaviour-pattern graph is also constrained to be linear in $m$:

$$|E_b| = O(m), \qquad T_{\text{beh-aff}} = O(K\,|E_b|) = O(Km). \tag{15}$$

Combining the two parts, the total MPGA cost per epoch is

$$T_{\text{MPGA}} = O(Km). \tag{16}$$

### A.6.5 GCN over the Unified Graph

The unified graph $G^\star$ has adjacency $A^\star = \lambda_1 A + \lambda_2 A_e + \lambda_3 A_b$, where $A$, $A_e$, and $A_b$ are the row-normalised adjacencies of $G$, $G_e$, and $G_b$, respectively. Because both auxiliary graphs are sparsified, the number of edges in $G^\star$ satisfies

$$m^\star = |E^\star| = |E| + |E_e| + |E_b| = O(m). \tag{17}$$

A standard implementation of an $L$-layer GCN over a sparse graph with $m^\star$ edges and $d$-dimensional node features has complexity

$$T_{\text{GCN}} = O(L\,m^\star d) = O(L\,md). \tag{18}$$

### A.6.6 Total Per-epoch Time Complexity

Summing the per-epoch costs of the components above, we obtain

$$T_{\text{epoch}} = T_{\text{MAPE}} + T_{\text{MPGA}} + T_{\text{GCN}} = O(nK\bar{L}) + O(Km) + O(L\,md). \tag{19}$$

Thus, one training epoch of MAPGA is linear in both the number of nodes $n$ and the number of edges $m$, with constant factors determined by the number of views $K$, the average number of partition rules per view $\bar{L}$, and the depth $L$ and width $d$ of the GCN backbone. In our experiments we adopt moderate hyperparameters (e.g., $K$ in the range 40–50 and small $L$), so the additional overhead over a vanilla GCN-based detector remains a modest constant factor in practice.

### A.6.7 SPACE COMPLEXITY

MAPGA also has scalable memory usage. Each view-specific codebook $E_k \in \mathbb{R}^{2^{L_k} \times d_0}$ contributes $O(2^{L_k} d_0)$ parameters; since the trees are shallow in practice, $L_k$ is small and the total codebook size $\sum_{k=1}^{K} 2^{L_k} d_0$ remains moderate. The unified adjacency $A^\star$ is stored in sparse form with $m^\star = O(m)$ edges, yielding $O(m)$ memory for graph structure, as in standard sparse GNNs. Finally, the node-embedding matrix has size $n \times d = n \times K d_0$, which is linear in the number of nodes. Overall, the space complexity of MAPGA is linear in $|V|$ and $|E|$, plus a manageable codebook term controlled by $K$, $\bar{L}$, and $d_0$.

### A.7 THEORETICAL ANALYSIS OF MANIFOLD CONNECTEDNESS AND MAPE

This section provides a formal analysis of the relationship between manifold connectedness and heterogeneous tabular node attributes, explains why fragmented manifolds hinder optimization of deep models, and shows how the proposed Multi-view Adaptive Partition Encoder (MAPE) induces approximate manifold connectedness in the representation space. Throughout, we adopt the same notation as in the main paper and briefly restate key definitions for completeness.

#### A.7.1 PRELIMINARIES AND NOTATION

Let $G = (V, E)$ be a directed graph with node set $V = \{1, \ldots, |V|\}$ and edge set $E \subseteq V \times V$. Each node $m \in V$ is associated with a feature vector $F^{(m)} = (f_1^{(m)}, \ldots, f_N^{(m)}) \in \mathbb{R}^N$, where different coordinates may correspond to semantically heterogeneous attributes (e.g., age, income, transaction counts, one-hot IDs).

**Definition A.1** (Feature distribution and data manifold). *Let $X$ be a random vector in $\mathbb{R}^N$ with distribution $P_X$ representing the population of node features. The data manifold (or support set) of $P_X$ is*

$$\mathcal{M} := \mathrm{supp}(P_X) = \{x \in \mathbb{R}^N : P_X(B_\varepsilon(x)) > 0 \text{ for all } \varepsilon > 0\},$$

*where $B_\varepsilon(x)$ denotes the open ball of radius $\varepsilon$ around $x$ in the ambient Euclidean space.*

**Definition A.2** (Manifold connectedness and fragmentation). *We say that $\mathcal{M}$ has manifold connectedness if it is path-connected in the subspace topology induced by $\mathbb{R}^N$, i.e., for any $x, x' \in \mathcal{M}$ there exists a continuous curve $\gamma : [0,1] \to \mathcal{M}$ with $\gamma(0) = x$ and $\gamma(1) = x'$. If $\mathcal{M}$ can be written as a finite union of pairwise disjoint, non-empty closed sets*

$$\mathcal{M} = \bigcup_{j=1}^{K} \mathcal{M}_j, \quad \mathcal{M}_i \cap \mathcal{M}_j = \emptyset \text{ for } i \neq j,$$

*such that for some $\Delta > 0$*

$$\mathrm{dist}(\mathcal{M}_i, \mathcal{M}_j) := \inf_{x \in \mathcal{M}_i,\, x' \in \mathcal{M}_j} \|x - x'\|_2 \geq \Delta \quad \forall i \neq j,$$

*then we say that $\mathcal{M}$ is fragmented and that manifold connectedness is absent.*

For a finite sample $\{F^{(m)}\}_{m \in V}$, we also consider the empirical $k$-nearest-neighbor (kNN) graph in feature space.

**Definition A.3** (Empirical $\varepsilon$-connectedness of the feature manifold). *Given samples $\{F^{(m)}\}_{m \in V}$ in $\mathbb{R}^N$ and a radius $\varepsilon > 0$, define the feature-neighborhood graph $G_\varepsilon = (V, E_\varepsilon)$ by $(m, m') \in E_\varepsilon$ iff $\|F^{(m)} - F^{(m')}\|_2 \leq \varepsilon$. We say the sample is $\varepsilon$-connected in feature space if $G_\varepsilon$ contains a single giant connected component containing $1 - o(1)$ fraction of vertices.*

*This graph is constructed purely from pairwise distances in feature space and is independent of the original relational edges $E$ of the graph.*

Empirically, Figure 1 of the main paper shows that audio, image, and text features yield a single, dense cluster in a low-dimensional t-SNE projection, whereas tabular features break into many isolated micro-clusters separated by large voids, indicating the absence of manifold connectedness in the tabular setting.

### A.7.2 Feature Heterogeneity and Manifold Fragmentation

We now formalize how heterogeneous tabular attributes can induce fragmented supports and why manifold connectedness is closely tied to the learnability of feature interactions.

**Heterogeneous tabular features** Let the index set of feature dimensions $\{1, \ldots, N\}$ be partitioned into $H$ semantically distinct groups (e.g., demographic, behavioral, structural):

$$\{1, \ldots, N\} = G_1 \dot{\cup} \ldots \dot{\cup} G_H, \quad G_h \neq \emptyset.$$

For $x \in \mathbb{R}^N$, denote by $x_{G_h}$ the subvector of coordinates in group $G_h$.

**Definition A.4** (Heterogeneous feature generative model). *We say that tabular features are hetero-geneous if there exist latent discrete variables $(C_1, \ldots, C_H)$, with finite supports, such that*

1. *$P_X$ can be written as a mixture*

$$P_X(x) = \sum_{c_1, \ldots, c_H} \pi_{c_1, \ldots, c_H} P(x_{G_1} \mid C_1 = c_1) \cdots P(x_{G_H} \mid C_H = c_H),$$

   *where $\pi_{c_1, \ldots, c_H} \geq 0$ and $\sum_{c_1, \ldots, c_H} \pi_{c_1, \ldots, c_H} = 1$;*

2. *for many pairs of configurations $(c_1, \ldots, c_H)$ and $(c'_1, \ldots, c'_H)$, the corresponding high-density regions in $\mathbb{R}^N$ are well-separated, e.g., the sets*

$$R_{c_1, \ldots, c_H} := \{x \in \mathbb{R}^N : p(x_{G_h} \mid C_h = c_h) \text{ is large for each } h\}$$

   *satisfy $\mathrm{dist}(R_{c_1, \ldots, c_H}, R_{c'_1, \ldots, c'_H}) \geq \Delta$ for a positive fraction of configuration pairs.*

Intuitively, each configuration $(c_1, \ldots, c_H)$ corresponds to a combination of discrete semantic regimes (e.g., age group, income bracket, transaction profile), and the joint distribution is concentrated near a finite set of such combinations. This formalizes the intuition that, for instance, age and income are both scalar-valued but arise from different generative factors and only appear in certain combinations in practice.

**Heterogeneity implies fragmented supports** Under mild conditions, heterogeneous generation induces a fragmented support.

**Assumption A.5** (Separated semantic configurations). *There exist configurations $\mathcal{C} = \{c^{(1)}, \ldots, c^{(K)}\}$ of $(C_1, \ldots, C_H)$ and closed sets $\mathcal{M}_j \subset \mathbb{R}^N$ for $j = 1, \ldots, K$ such that*

1. *$P_X$ is supported on $\mathcal{M} = \bigcup_{j=1}^K \mathcal{M}_j$;*

2. *$\mathcal{M}_i \cap \mathcal{M}_j = \emptyset$ for $i \neq j$;*

3. *for some $\Delta > 0$, $\mathrm{dist}(\mathcal{M}_i, \mathcal{M}_j) \geq \Delta$ for all $i \neq j$.*

Assumption A.5 states that different high-probability semantic regimes occupy disjoint, well-separated regions in feature space, as is typical for real-world tabular data with hard thresholds and business rules.

**Proposition A.6** (Heterogeneity induces manifold fragmentation). *If the feature distribution is het-erogeneous in the sense of Definition A.4 and satisfies Assumption A.5, then the data manifold $\mathcal{M}$ is fragmented in the sense of Definition A.2, i.e., $\mathcal{M} = \bigcup_{j=1}^K \mathcal{M}_j$ with pairwise distance at least $\Delta > 0$ and at least two connected components.*

*Proof.* By Assumption A.5, any two distinct sets $\mathcal{M}_i, \mathcal{M}_j$ are closed, disjoint, and satisfy $\mathrm{dist}(\mathcal{M}_i, \mathcal{M}_j) \geq \Delta > 0$. Suppose by contradiction that $\mathcal{M}$ is path-connected. Then for any $x \in \mathcal{M}_i$ and $x' \in \mathcal{M}_j$ there exists a continuous path $\gamma : [0, 1] \to \mathcal{M}$ with $\gamma(0) = x$, $\gamma(1) = x'$. By continuity, the sets $T_i = \{t : \gamma(t) \in \mathcal{M}_i\}$ and $T_j = \{t : \gamma(t) \in \mathcal{M}_j\}$ are closed and non-empty subsets of $[0, 1]$. Since $\mathcal{M}_i$ and $\mathcal{M}_j$ are disjoint and their distance is strictly positive, there must exist $t^\star$ such that $\gamma(t^\star) \notin \mathcal{M}_i \cup \mathcal{M}_j$ and $\gamma(t^\star) \notin \mathcal{M}_\ell$ for all $\ell$, because otherwise the path could not cross from one closed, separated set to another. But then $\gamma(t^\star) \notin \mathcal{M}$, contradicting the assumption $\gamma([0, 1]) \subseteq \mathcal{M}$. Hence $\mathcal{M}$ cannot be path-connected, and it is fragmented. $\square$

**Remark A.7.** *In finite samples, Proposition A.6 corresponds to the empirical observation that tabular features cluster into many isolated micro-clusters in t-SNE space (Figure 1 of the main paper), with large voids between clusters. By contrast, perceptual features (audio, image, text) are known to form a single, relatively smooth manifold with coherent neighborhoods (Bengio et al., 2013).*

**Manifold connectedness and learnable interactions** We next relate manifold connectedness to the learnability of feature interactions by locally smooth models.

**Definition A.8** (Locally Lipschitz label function). *Let $(\mathcal{M}, d_{\mathcal{M}})$ be a metric space, where $d_{\mathcal{M}}$ denotes the intrinsic (e.g., geodesic) distance on the manifold. A function $f^{\star} : \mathcal{M} \to \mathbb{R}$ is said to be locally $L_0$-Lipschitz if for every $x \in \mathcal{M}$ there exists $r_x > 0$ such that*

$$\left|f^{\star}(x) - f^{\star}(x')\right| \le L_0 \, d_{\mathcal{M}}(x, x') \quad \text{whenever } d_{\mathcal{M}}(x, x') \le r_x.$$

Such a condition underpins manifold regularization and semi-supervised learning methods, where the assumption that labels vary smoothly along the data manifold is exploited for sample-efficient learning (Belkin et al., 2006; Bengio et al., 2013).

**Definition A.9** (Interaction learnability under local smoothness). *Let $\mathcal{F}_L$ denote a class of neural networks $f_\theta : \mathbb{R}^N \to \mathbb{R}$ that are globally $L$-Lipschitz in the ambient Euclidean norm: $\|f_\theta(x) - f_\theta(x')\| \le L\|x - x'\|_2$ for all $x, x'$. We say that feature interactions on $\mathcal{M}$ are locally learnable by $\mathcal{F}_L$ if for any locally Lipschitz $f^{\star} : \mathcal{M} \to \mathbb{R}$ and any $\varepsilon > 0$ there exists $f_\theta \in \mathcal{F}_L$ such that*

$$\mathbb{E}_{X \sim P_X}\left[|f_\theta(X) - f^{\star}(X)|\right] \le \varepsilon,$$

*with approximation error controlled predominantly by local properties of $\mathcal{M}$ (dimension, curvature) rather than by the number of disconnected components.*

A single connected manifold with bounded curvature and dimension typically admits such approximation by Lipschitz neural networks with moderate complexity (Yarotsky, 2017; Barron & Klusowski, 2019).

**Proposition A.10** (Fragmentation increases effective complexity). *Let $\mathcal{M} = \bigcup_{j=1}^{K} \mathcal{M}_j$ be fragmented as in Definition A.2, with $\operatorname{dist}(\mathcal{M}_i, \mathcal{M}_j) \ge \Delta > 0$ for all $i \neq j$. Suppose $f^{\star}$ is piecewise constant with values $a_j \in \mathbb{R}$ on each component $\mathcal{M}_j$. Then any globally $L$-Lipschitz function $f : \mathbb{R}^N \to \mathbb{R}$ that satisfies $|f(x) - f^{\star}(x)| \le \epsilon$ for all $x \in \mathcal{M}$ must obey*

$$L \ge \frac{\max_{i \neq j} |a_i - a_j| - 2\epsilon}{\Delta}.$$

*In particular, if $\epsilon$ is small and the differences $|a_i - a_j|$ are $O(1)$, then $L$ must be at least $\Omega(1/\Delta)$.*

*Proof.* Fix $i \neq j$ and choose $x \in \mathcal{M}_i$, $x' \in \mathcal{M}_j$ such that $\|x - x'\|_2 = \operatorname{dist}(\mathcal{M}_i, \mathcal{M}_j) \ge \Delta$. Since $f$ is $L$-Lipschitz, we have

$$|f(x) - f(x')| \le L\|x - x'\|_2 \le L\Delta.$$

On the other hand,

$$\left|f(x) - f(x')\right| \ge \left|a_i - a_j\right| - |f(x) - a_i| - |f(x') - a_j| \ge \left|a_i - a_j\right| - 2\epsilon.$$

Combining,

$$L\Delta \ge \left|a_i - a_j\right| - 2\epsilon,$$

so

$$L \ge \frac{\left|a_i - a_j\right| - 2\epsilon}{\Delta}.$$

Taking the maximum over all $i \neq j$ yields the claim. □

Proposition A.10 shows that even for extremely simple target functions (piecewise constants on components), strong fragmentation forces any globally Lipschitz model to have large Lipschitz constant, thereby reducing the effectiveness of the local-smoothness prior. In practice, weight decay and spectral regularization tend to keep the Lipschitz constant moderate, causing underfitting across fragmented manifolds.

### A.7.3 DEEP NEURAL NETWORKS ON FRAGMENTED MANIFOLDS

We now analyze how manifold fragmentation and heterogeneous coordinates translate into optimization difficulties for gradient-based deep models.

**Lipschitz neural networks and local smoothness** A standard feed-forward neural network with ReLU or other piecewise linear activations can be written as

$$f_\theta(x) = W_L \sigma\big(W_{L-1}\sigma(\cdots \sigma(W_1 x + b_1) + b_{L-1})\big) + b_L,$$

where $\sigma$ is 1-Lipschitz (e.g., ReLU, GELU) and each $W_\ell$ is a weight matrix. Under mild assumptions (bounded operator norms), $f_\theta$ is globally Lipschitz with constant $L_\theta \leq \prod_{\ell=1}^{L} \|W_\ell\|_{2\to2}$ (Goodfellow, 2016). This Lipschitz property encodes a local smoothness prior: nearby inputs should map to nearby outputs.

In graph neural networks (GNNs), linear layers are combined with aggregation over neighbors; the resulting architectures are also Lipschitz under appropriate normalizations of the adjacency matrix (Kipf & Welling, 2016; Wu et al., 2019).

**Fragmentation and gradient variance** Consider empirical risk minimization

$$\mathcal{L}(\theta) = \mathbb{E}_{(X,Y)\sim P_{XY}}\big[\ell(f_\theta(X), Y)\big] \approx \frac{1}{|V|} \sum_{m\in V} \ell(f_\theta(F^{(m)}), y^{(m)}),$$

with a smooth loss $\ell$ (e.g., cross-entropy).

Assume the data manifold decomposes as in Proposition A.6: $\mathcal{M} = \bigcup_{j=1}^{K} \mathcal{M}_j$, and let $\pi_j$ denote the mass of component $\mathcal{M}_j$: $\pi_j = \mathbb{P}[X \in \mathcal{M}_j]$. We can decompose the population loss as

$$\mathcal{L}(\theta) = \sum_{j=1}^{K} \pi_j \, \mathcal{L}_j(\theta), \quad \mathcal{L}_j(\theta) = \mathbb{E}\big[\ell(f_\theta(X), Y) \mid X \in \mathcal{M}_j\big].$$

**Lemma A.11** (Cluster-wise gradient decomposition). *Let $g(\theta) = \nabla_\theta \mathcal{L}(\theta)$ and $g_j(\theta) = \nabla_\theta \mathcal{L}_j(\theta)$. Then*

$$g(\theta) = \sum_{j=1}^{K} \pi_j \, g_j(\theta).$$

*For a single-sample stochastic gradient estimator $\widehat{g}(\theta)$ obtained by drawing $(X,Y)$ from $P_{XY}$,*

$$\mathrm{Var}[\widehat{g}(\theta)] = \sum_{j=1}^{K} \pi_j \big\|g_j(\theta) - g(\theta)\big\|_2^2 + \mathbb{E}[\mathrm{Var}(\nabla_\theta \ell(f_\theta(X), Y) \mid X)].$$

*Proof.* The equality $g(\theta) = \sum_j \pi_j g_j(\theta)$ follows by linearity of differentiation and the law of total expectation. The variance identity is obtained via the law of total variance applied to the random gradient $\nabla_\theta \ell(f_\theta(X), Y)$ with respect to the partition $\{X \in \mathcal{M}_j\}_{j=1}^{K}$. □

**Proposition A.12** (Fragmentation inflates gradient variance). *Suppose there exist components $i, j$ such that $\|g_i(\theta) - g_j(\theta)\|_2 \geq \delta_g > 0$ at some $\theta$. Then*

$$\mathrm{Var}[\widehat{g}(\theta)] \ \geq \ \frac{1}{2}\pi_i\pi_j\delta_g^2,$$

*ignoring within-cluster conditional variance. In particular, as the gradients $g_j(\theta)$ diverge across components, the variance of stochastic gradients grows, slowing the convergence of stochastic gradient descent (SGD) (Bottou et al., 2018).*

*Proof.* By Lemma A.11, the between-cluster part of the variance is

$$\sum_{j=1}^{K} \pi_j \big\|g_j(\theta) - g(\theta)\big\|_2^2.$$

For any two vectors $u, v$ and weights $\alpha, \beta \geq 0$ with $\alpha + \beta \leq 1$, the variance of a two-component mixture satisfies

$$\alpha \|u - m\|_2^2 + \beta \|v - m\|_2^2 \geq \frac{\alpha\beta}{\alpha + \beta} \|u - v\|_2^2,$$

where $m = (\alpha u + \beta v)/(\alpha + \beta)$. Applying this inequality with $u = g_i(\theta)$, $v = g_j(\theta)$, $\alpha = \pi_i$, $\beta = \pi_j$ and observing that the remainder of the mixture only increases variance yields

$$\mathrm{Var}[\widehat{g}(\theta)] \geq \frac{\pi_i \pi_j}{\pi_i + \pi_j} \|g_i(\theta) - g_j(\theta)\|_2^2 \geq \frac{1}{2} \pi_i \pi_j \delta_g^2,$$

since $\pi_i + \pi_j \leq 1$. $\qquad\square$

Combined with standard SGD convergence results for smooth (possibly non-convex) objectives (Bottou et al., 2018; Ghadimi & Lan, 2013), Proposition A.12 shows that fragmentation exacerbates gradient noise, particularly when clusters are imbalanced (e.g., rare anomalies) and gradients differ substantially between them.

**Heterogeneous coordinates and ill-conditioning**   Heterogeneous feature groups further aggravate optimization by inducing poorly conditioned Hessians.

Consider a simple linear model $f_\theta(x) = \theta^\top x$ with squared loss on data $(x, y) \in \mathbb{R}^N \times \mathbb{R}$, leading to population risk

$$\mathcal{L}(\theta) = \frac{1}{2} \mathbb{E}\big[(\theta^\top X - Y)^2\big].$$

The Hessian is the covariance matrix of features, $H = \nabla^2 \mathcal{L}(\theta) = \Sigma := \mathbb{E}[XX^\top]$. The condition number $\kappa(H) = \lambda_{\max}(\Sigma)/\lambda_{\min}(\Sigma)$ governs the convergence rate of gradient descent (Nesterov, 2013).

In heterogeneous tabular data, different feature groups $G_h$ often have vastly different variances and correlations (e.g., age in years, income in thousands of dollars, sparse one-hot IDs). As a result, $\Sigma$ becomes ill-conditioned and the effective step size needed for stable convergence is dominated by the largest eigenvalue of $\Sigma$, causing very slow progress along small-variance directions.

Although deep networks can partially re-scale features through learned weights, they face exactly the same difficulty during early training: linear layers must learn appropriate rescalings from noisy gradients computed on fragmented data, which is particularly challenging when samples occupy only a small subset of possible feature combinations.

### A.7.4   MAPE AS A MANIFOLD-CONNECTEDNESS INDUCING MAP

We now analyze how the proposed MAPE encoder maps the fragmented feature manifold $\mathcal{M}$ to a representation space $\mathcal{Z}$ with enhanced connectedness and local smoothness.

**MAPE recap and formalization**   We recall the construction in Section 3.1 of the main paper. For each node $m$, the raw feature vector is $F^{(m)} \in \mathbb{R}^N$. MAPE instantiates $K$ views $\mathcal{V} = \{v_1, \ldots, v_K\}$. For each view $k$, we define $L_k$ binary decision rules

$$R_k = \{r_k^1, \ldots, r_k^{L_k}\}, \quad r_k^\ell = (n_k^\ell, s_k^\ell),$$

where $n_k^\ell \in \{1, \ldots, N\}$ selects a feature coordinate and $s_k^\ell \in \mathbb{R}$ is a learnable split point.

For node $m$ and view $k$, the $\ell$-th decision rule outputs

$$b_{m,k}^\ell = H\big(f_{n_k^\ell}^{(m)} - s_k^\ell\big) \in \{0, 1\},$$

where $H(\cdot)$ is a Heaviside step function (with a smooth surrogate used during training). Stacking all rule outputs yields a binary code

$$c_k^{(m)} = (b_{m,k}^1, \ldots, b_{m,k}^{L_k})^\top \in \{0, 1\}^{L_k}.$$

Geometrically, these rules realize an axis-aligned hyper-rectangular partition of the subspace spanned by $\{f_{n_k^\ell}\}_{\ell=1}^{L_k}$. Each binary code $c_k$ identifies a hyper-rectangular cell (or discrete subspace)

$$S_k(c_k) = \left\{ x \in \mathbb{R}^N : H(x_{n_k^\ell} - s_k^\ell) = c_k^\ell, \ \ell = 1, \ldots, L_k \right\}.$$

For each view $k$, we allocate a trainable dictionary $E_k \in \mathbb{R}^{2^{L_k} \times d_0}$ and define an embedding lookup function $\phi_k : \{0,1\}^{L_k} \to \mathbb{R}^{d_0}$ via table lookup:

$$z_k^{(m)} = \phi_k(c_k^{(m)}) = E_k[\mathrm{idx}(c_k^{(m)})],$$

where $\mathrm{idx}(c_k)$ maps $c_k$ to an integer index in $\{1, \ldots, 2^{L_k}\}$. The final representation for node $m$ is

$$z^{(m)} = \left[ z_1^{(m)} \| \cdots \| z_K^{(m)} \right] \in \mathbb{R}^d, \quad d = K d_0.$$

Let $\mathcal{Z}$ denote the image of $\mathcal{M}$ under the MAPE map,

$$\Phi : \mathcal{M} \to \mathcal{Z}, \quad \Phi(x) = z(x).$$

**A subspace-level connectivity structure**   We next show that MAPE induces a combinatorial connectivity structure on the discrete subspaces, which then translates into improved connectedness among embedded points.

For a fixed view $k$, the $2^{L_k}$ codes define $2^{L_k}$ cells $S_k(c)$, and we can build a *subspace graph* connecting cells that differ by a single bit.

**Definition A.13** (Per-view subspace graph). *For view $k$, define a graph $G_k^{sub} = (V_k^{sub}, E_k^{sub})$ with vertex set $V_k^{sub} = \{0,1\}^{L_k}$ and edge set*

$$E_k^{sub} = \left\{ (c, c') : c, c' \in \{0,1\}^{L_k}, \ \mathrm{Ham}(c, c') = 1 \right\},$$

*where $\mathrm{Ham}(\cdot, \cdot)$ denotes Hamming distance.*

Each vertex $c \in V_k^{\text{sub}}$ corresponds to a hyper-rectangular cell $S_k(c)$, and an edge connects two cells that differ by a single threshold crossing in one coordinate.

**Lemma A.14** (Connectivity of the subspace graph). *For each view $k$, the subspace graph $G_k^{sub}$ is connected.*

*Proof.* The graph $G_k^{\text{sub}}$ is the $L_k$-dimensional hypercube. It is well known that the hypercube graph is connected: given any two codes $c$ and $c'$, flipping differing bits one by one yields a path of length $\mathrm{Ham}(c, c')$ connecting them. □

While Lemma A.14 is purely combinatorial, we now connect it to the actual data distribution and the embeddings.

**Assumption A.15** (Non-empty cells). *For each view $k$, there exists a subset $\mathcal{C}_k \subseteq \{0,1\}^{L_k}$ such that each cell $S_k(c)$ with $c \in \mathcal{C}_k$ has positive probability under $P_X$, i.e., $P_X(S_k(c)) > 0$, and $P_X\left(\bigcup_{c \in \mathcal{C}_k} S_k(c)\right) = 1$.*

Assumption A.15 says that MAPE partitions do not produce degenerate empty cells at convergence; empirical histograms of code frequencies confirm this in practice.

**Definition A.16** (MAPE-induced neighbor graph). *Let $\{x^{(m)}\}_{m \in V}$ be the raw features and $\{z^{(m)}\}_{m \in V}$ be their MAPE embeddings. Define a graph $G_Z = (V, E_Z)$ where $(m, m') \in E_Z$ if and only if there exists at least one view $k$ such that $c_k^{(m)} = c_k^{(m')}$, i.e., $x^{(m)}$ and $x^{(m')}$ fall into the same cell $S_k(c)$ in some view.*

Edges in $G_Z$ therefore indicate that two nodes share a discrete subspace in at least one view, reflecting a high-order joint condition on raw features (e.g., "age $< 30$ and income $> 1000$").

**Probabilistic connectivity of the neighbor graph**    We now study the connectivity of $G_Z$ under a mild probabilistic model for the partition rules. For theoretical tractability, we consider a simplified setting in which views are randomized independently.

**Assumption A.17** (Randomized views with bounded per-cell mass).  *For each view $k$:*

> 1. *The set of rules $R_k$ is generated by a randomized procedure (e.g., random feature selection and thresholding) independent across views;*
>
> 2. *There exist constants $0 < \rho_{\min} \leq \rho_{\max} < 1$ such that for all $c \in \mathcal{C}_k$,*
>
> $$\rho_{\min} \leq P_X\big(X \in S_k(c)\big) \leq \rho_{\max};$$
>
> 3. *For i.i.d. draws $X$, $X'$ from $P_X$, the indicator $\mathbf{1}\{c_k(X) = c_k(X')\}$ has probability*
>
> $$q_k := \mathbb{P}\big[c_k(X) = c_k(X')\big] = \sum_c P_X(S_k(c))^2 \geq 2^{L_k}\rho_{\min}^2 =: q_{\min} > 0.$$

Assumption A.17 can be viewed as a lower-bound model for the overlap induced by MAPE: in practice, tree-initialized and task-refined partitions increase $q_k$ for semantically similar nodes relative to random partitions.

Let $M = |V|$ denote the number of nodes. For each pair of distinct indices $(m, m')$ and each view $k$, define
$$A_k(m, m') = \mathbf{1}\big\{c_k^{(m)} = c_k^{(m')}\big\}.$$

Under Assumption A.17, conditioning on $X^{(m)}$ and $X^{(m')}$, the random variables $\{A_k(m, m')\}_{k=1}^K$ are independent Bernoulli with success probabilities at least $q_{\min}$.

Thus, the probability that $(m, m')$ share a cell in at least one view is
$$p_{m,m'} := \mathbb{P}\big[(m, m') \in E_Z\big] = 1 - \mathbb{P}\big[A_k(m, m') = 0 \ \forall k\big] \geq 1 - (1 - q_{\min})^K.$$

**Lemma A.18** (Lower-bounded edge probability).  *Define*
$$p^\star := 1 - (1 - q_{\min})^K.$$
*Then for all distinct node pairs $(m, m')$, we have*
$$\mathbb{P}\big[(m, m') \in E_Z\big] \geq p^\star.$$

We now relate $G_Z$ to an Erdős–Rényi graph $G_{\mathrm{ER}}(M, p^\star)$.

**Theorem A.19** (Probabilistic connectivity of $G_Z$).  *Under Assumption A.17, suppose*
$$p^\star := 1 - (1 - q_{\min})^K \geq \frac{(1 + \varepsilon)\log M}{M}$$

*for some fixed $\varepsilon > 0$. Then the MAPE-induced neighbor graph $G_Z$ is connected with probability tending to 1 as $M \to \infty$. In particular, it suffices to choose the number of views $K$ so that*
$$K \geq \frac{\log\big(1 - (1 + \varepsilon)\frac{\log M}{M}\big)}{\log(1 - q_{\min})} \approx \frac{\log M}{q_{\min}}.$$

*Proof.*  Consider the Erdős–Rényi random graph $G_{\mathrm{ER}}(M, p^\star)$, where each edge between distinct vertices is present independently with probability $p^\star$. It is a classical result (ERDdS & R&wi, 1959; Bollobás, 2011) that if $p^\star \geq (1 + \varepsilon)\frac{\log M}{M}$, then $G_{\mathrm{ER}}(M, p^\star)$ is connected with probability approaching 1 as $M \to \infty$.

We claim that we can couple the edge sets of $G_Z$ and $G_{\mathrm{ER}}(M, p^\star)$ so that $E_{\mathrm{ER}} \subseteq E_Z$ almost surely. Indeed, for each pair $(m, m')$, draw a uniform random variable $U_{m,m'} \sim \mathrm{Unif}[0, 1]$ independent of all other randomness. Let
$$\mathbf{1}\big[(m, m') \in E_{\mathrm{ER}}\big] = \mathbf{1}\big[U_{m,m'} \leq p^\star\big],$$

and let

$$\mathbf{1}\big[(m, m') \in E_Z\big] = \mathbf{1}\big[U_{m,m'} \le p_{m,m'}\big].$$

By Lemma A.18, $p_{m,m'} \ge p^\star$, hence $\mathbf{1}[(m, m') \in E_{\mathrm{ER}}] \le \mathbf{1}[(m, m') \in E_Z]$ almost surely for each pair. Therefore $E_{\mathrm{ER}} \subseteq E_Z$ almost surely. Connectivity is a monotone (increasing) graph property, so

$$\mathbb{P}[G_Z \text{ connected}] \ \ge \ \mathbb{P}[G_{\mathrm{ER}}(M, p^\star) \text{ connected}] \to 1 \quad \text{as } M \to \infty.$$

□

Theorem A.19 shows that, under mild probabilistic assumptions and a sufficient number of views $K$, the neighbor graph $G_Z$ induced by MAPE is connected with high probability, even when the original feature manifold $\mathcal{M}$ is fragmented into many components.

**Remark A.20.** *In the actual algorithm, views are initialized from shallow decision trees and then refined via back-propagation using the downstream task loss. This data-adaptive mechanism typically increases the overlap probabilities $q_k$ for semantically similar nodes and decreases them for dissimilar nodes, which further enhances the effective connectivity among normal nodes while separating anomalies, as evidenced by the t-SNE visualizations in Figure 2 of the main paper.*

**Local smoothness in the MAPE representation space**  We now analyze the geometry of the embedded representations $z^{(m)} \in \mathcal{Z}$.

**Assumption A.21** (Bounded embedding norms)**.** *For each view $k$, there exists $B > 0$ such that for all rows $v \in \{1, \ldots, 2^{L_k}\}$ in $E_k$,*

$$\|E_k[v, :]\|_2 \le B.$$

Assumption A.21 is natural under standard weight-decay or norm-regularization schemes.

**Lemma A.22** (Distance bound for shared subspaces)**.** *Let $m, m' \in V$ and suppose there is a set of views $\mathcal{K}_{share} \subseteq \{1, \ldots, K\}$ such that $c_k^{(m)} = c_k^{(m')}$ for all $k \in \mathcal{K}_{share}$. Then under Assumption A.21,*

$$\|z^{(m)} - z^{(m')}\|_2 \le 2B \sqrt{K - |\mathcal{K}_{share}|}.$$

*In particular, if $(m, m') \in E_Z$ (i.e., they share at least one view), then*

$$\|z^{(m)} - z^{(m')}\|_2 \le 2B\sqrt{K - 1}.$$

*Proof.* We have

$$z^{(m)} - z^{(m')} = \big[z_1^{(m)} - z_1^{(m')}\| \cdots \|z_K^{(m)} - z_K^{(m')}\big].$$

For any view $k$ with $c_k^{(m)} = c_k^{(m')}$, we have $z_k^{(m)} = z_k^{(m')}$ by definition of $\phi_k$, so the contribution to the squared distance from this view is zero. For views where $c_k^{(m)} \ne c_k^{(m')}$, we have

$$\|z_k^{(m)} - z_k^{(m')}\|_2 \le \|z_k^{(m)}\|_2 + \|z_k^{(m')}\|_2 \le 2B,$$

by Assumption A.21. Let $\mathcal{K}_{\mathrm{diff}} = \{1, \ldots, K\} \setminus \mathcal{K}_{\mathrm{share}}$ denote the set of views where the codes differ. Then

$$\|z^{(m)} - z^{(m')}\|_2^2 = \sum_{k \in \mathcal{K}_{\mathrm{diff}}} \|z_k^{(m)} - z_k^{(m')}\|_2^2 \le \sum_{k \in \mathcal{K}_{\mathrm{diff}}} (2B)^2 = 4B^2 |\mathcal{K}_{\mathrm{diff}}| = 4B^2 (K - |\mathcal{K}_{\mathrm{share}}|),$$

so the claimed bound follows by taking square roots. If $(m, m') \in E_Z$, then $|\mathcal{K}_{\mathrm{share}}| \ge 1$, giving $\|z^{(m)} - z^{(m')}\|_2 \le 2B\sqrt{K - 1}$. □

Lemma A.22 implies that, along edges of the neighbor graph $G_Z$, the embedding distances are uniformly bounded. Because $G_Z$ is connected with high probability (Theorem A.19), we can view $\mathcal{Z}$ as a graph metric space whose geodesic distances along $G_Z$ are controlled by bounded-length steps. This yields an approximate manifold structure on $\mathcal{Z}$.

**Definition A.23** (MAPE representation manifold). *Let $G_Z = (V, E_Z)$ be the neighbor graph on embedded points $\{z^{(m)}\}_{m \in V}$. Define the graph-based geodesic distance*

$$d_{\mathcal{Z}}(m, m') = \inf\Big\{ \sum_{t=0}^{T-1} \|z^{(m_t)} - z^{(m_{t+1})}\|_2 : (m_t, m_{t+1}) \in E_Z,\ m_0 = m,\ m_T = m' \Big\}.$$

*We call the metric space $(\mathcal{Z}, d_{\mathcal{Z}})$ the MAPE representation manifold.*

**Proposition A.24** (Approximate manifold connectedness in $\mathcal{Z}$). *Under Assumptions A.17 and A.21, for sufficiently large $K$ we have with high probability:*

1. *$G_Z$ is connected (Theorem A.19);*

2. *along any shortest path in $G_Z$, each step has length at most $2B\sqrt{K-1}$ (Lemma A.22);*

3. *consequently, $(\mathcal{Z}, d_{\mathcal{Z}})$ is a connected metric space whose geodesics respect the combinatorial structure of shared subspaces.*

*Therefore, MAPE induces an approximately connected manifold structure in the representation space, even when the original feature manifold $\mathcal{M}$ is fragmented.*

**Improved compatibility with locally smooth models**    Finally, we argue that the composition of MAPE with a locally Lipschitz neural network restores the effectiveness of the local-smoothness prior.

Let $g_\theta : \mathbb{R}^d \to \mathbb{R}$ be a neural network (e.g., GNN, MLP) applied to $z^{(m)}$, and let $f_\theta = g_\theta \circ \Phi : \mathcal{M} \to \mathbb{R}$ be the overall predictor on raw features. Suppose $g_\theta$ is $L_g$-Lipschitz with respect to the Euclidean norm on $\mathbb{R}^d$.

**Theorem A.25** (MAPE restores local smoothness for deep models). *Assume:*

1. *The raw feature manifold $\mathcal{M}$ is fragmented as in Proposition A.6;*

2. *The target function $f^\star : \mathcal{M} \to \mathbb{R}$ is locally Lipschitz with respect to $d_{\mathcal{Z}}$ on $(\mathcal{Z}, d_{\mathcal{Z}})$ after MAPE, i.e., $f^\star(x) = \tilde{f}^\star(\Phi(x))$ for some $\tilde{f}^\star : \mathcal{Z} \to \mathbb{R}$ that is locally $L_0$-Lipschitz in $d_{\mathcal{Z}}$;*

3. *$G_Z$ is connected and Assumption A.21 holds.*

*Then there exists a neural network $g_\theta : \mathbb{R}^d \to \mathbb{R}$ with moderate Lipschitz constant $L_g$ such that $\mathbb{E}_{X \sim P_X}[|g_\theta(\Phi(X)) - f^\star(X)|]$ is arbitrarily small, with complexity and Lipschitz constant controlled mainly by local geometric quantities on $(\mathcal{Z}, d_{\mathcal{Z}})$ (e.g., an upper bound on degrees and local cluster sizes), rather than by the number of disconnected components of $\mathcal{M}$.*

*Sketch.* By assumption, there exists a locally $L_0$-Lipschitz function $\tilde{f}^\star$ on $(\mathcal{Z}, d_{\mathcal{Z}})$ such that $f^\star = \tilde{f}^\star \circ \Phi$. Since $(\mathcal{Z}, d_{\mathcal{Z}})$ is connected and built from bounded-length edges (Proposition A.24), existing approximation results for Lipschitz functions on finite metric spaces by neural networks (e.g., via 1-Lipschitz layers and linear readouts) imply that there exists $g_\theta$ with Lipschitz constant $L_g = O(L_0)$ that approximates $\tilde{f}^\star$ arbitrarily well on the finite set $\{z^{(m)}\}_{m \in V}$; see, e.g., Yarotsky (2017) and Anil et al. (2019) for constructions of networks approximating Lipschitz functions on discrete sets. Thus, the composition $f_\theta = g_\theta \circ \Phi$ can approximate $f^\star$ arbitrarily closely on the dataset with Lipschitz constant approximately $L_g$, which no longer needs to scale with the separation between components of $\mathcal{M}$, but only with local structure in $\mathcal{Z}$. $\qquad\square$

Theorem A.25 formalizes the central intuition of the main paper: MAPE remaps fragmented, heterogeneous tabular features into a representation space where a locally smooth neural model is well suited, thereby mitigating the mismatch between data geometry and model inductive bias. Empirically, this manifests as a single, well-connected manifold with smooth density transitions in t-SNE projections (Figure 2 in the main paper) and as improved convergence and anomaly-detection performance across benchmarks.

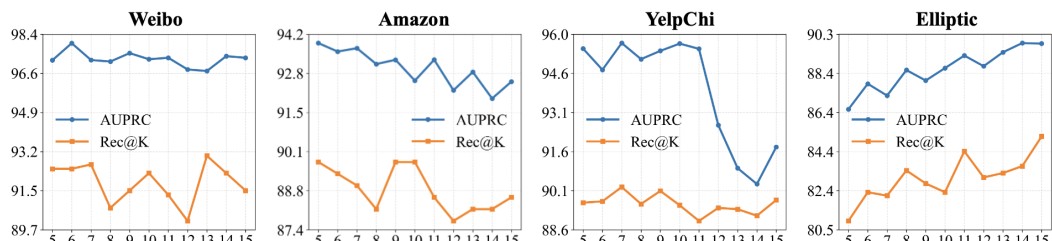

Figure 6: Impact of initialization tree depth on detection performance. AUPRC (solid blue) and Rec@K (dashed orange) versus maximum tree depth $D \in \{5, \dots, 15\}$ on four benchmarks. Performance is stable for moderate depths on Weibo, Amazon, and YelpChi, while Elliptic benefits from deeper trees, reflecting more complex feature interactions in its high-dimensional node attributes.

### A.7.5 SUMMARY

The theoretical developments in this section can be summarized as follows:

- Heterogeneous tabular features naturally induce fragmented manifolds, where the data support decomposes into multiple, well-separated components (Proposition A.6).

- On such fragmented manifolds, globally Lipschitz deep models must either adopt large Lipschitz constants or incur approximation error. Fragmentation inflates gradient variance and exacerbates ill-conditioning, slowing convergence and reducing robustness (Propositions A.10 and A.12).

- MAPE discretizes the feature space into multi-view subspaces and assigns learnable embeddings to shared subspaces, thereby constructing a neighbor graph $G_Z$ that is connected with high probability and whose edges correspond to bounded-length steps in embedding space (Theorem A.19, Lemma A.22, Proposition A.24).

- In the resulting representation manifold $(\mathcal{Z}, d_{\mathcal{Z}})$, a locally Lipschitz neural model can approximate the target function with moderate Lipschitz constant and improved compatibility with the local-smoothness prior (Theorem A.25).

Together, these results provide a principled explanation of how MAPE reconstructs approximate manifold connectedness for heterogeneous tabular node attributes and why this benefits downstream graph anomaly detection.

### A.8 IMPACT OF INITIALIZATION TREE DEPTH

MAPE uses a shallow CART tree per view to initialise the set of binary rules by converting each internal split node into one rule, after which all split thresholds are treated as trainable parameters and optimised end-to-end with the anomaly-detection loss. Consequently, the maximum depth $D$ only controls the number and granularity of initial partitions, while the final decision boundaries are determined by subsequent optimisation rather than by the fixed tree structure.

We vary $D \in \{5, \dots, 15\}$ on four representative benchmarks (Weibo, Amazon, YelpChi, Elliptic), keeping all other hyperparameters fixed, and report AUPRC and Rec@K in Fig. 7. On Weibo, Amazon, and YelpChi, both metrics remain nearly flat across a broad range of depths, with very deep trees sometimes causing a slight drop, likely because they over-fragment the feature space into tiny subregions with few samples. In contrast, Elliptic shows a gradual improvement as $D$ increases. Elliptic has high-dimensional node features and complex transactional patterns (166-dimensional attributes; see Table 3), so deeper trees provide a richer pool of initial rules that helps capture more intricate feature interactions before global optimisation. Overall, MAPGA is insensitive to the exact depth within a moderate range, and we therefore fix $D$ to a small constant in this stable region for all main experiments.

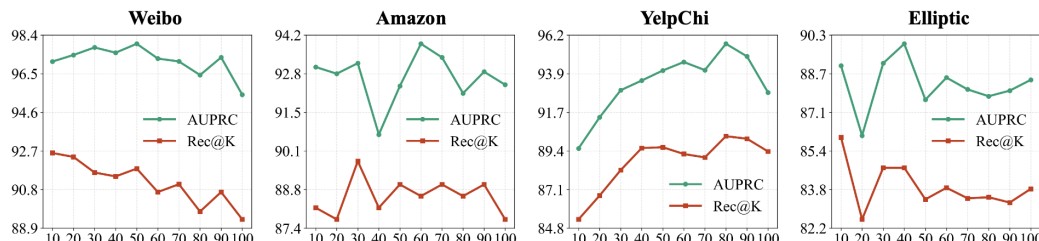

Figure 7: Effect of the surrogate slope $\alpha$ used in the sigmoid approximation of the Heaviside function. Detection performance (AUPRC and Rec@K) on four representative benchmarks (Weibo, Amazon, YelpChi, and Elliptic) as $\alpha$ varies from 10 to 100. MAPGA remains stable over a broad range of $\alpha$, and all main experiments adopt a single global setting $\alpha = 50$.

## A.9  EFFECT OF THE SURROGATE SLOPE $\alpha$

To study the sensitivity of MAPGA to the slope $\alpha$ in the sigmoid surrogate of the Heaviside function in Eq. (1), we vary $\alpha$ from 10 to 100 in steps of 10 and report AUPRC and Rec@K on four representative benchmarks (Weibo, Amazon, YelpChi, and Elliptic); the results are shown in Figure 7. Across all four datasets, both metrics are stable once $\alpha$ is moderately large. On Weibo and Elliptic, the curves form a broad plateau for $\alpha \geq 20$, with the best performance attained around $\alpha \in [40, 60]$ and only small fluctuations outside this range. On Amazon and YelpChi, increasing $\alpha$ from 10 to 40 yields a mild performance gain as the surrogate becomes sharper, after which both AUPRC and Rec@K saturate and vary only slightly.

These observations indicate that MAPGA is not overly sensitive to the exact choice of the surrogate slope, as long as $\alpha$ lies in a reasonable range that balances gradient smoothness and approximation fidelity. In all main experiments we therefore fix a single global value $\alpha = 50$ for every dataset and variant, which is located in the middle of the plateau region and avoids any per-dataset hyperparameter tuning.

