# OpenReview forum: "Multi‑view Adaptive Partitioning with Global Association for Graph Anomaly Detection"
_ICLR.cc/2026/Conference — Submitted to ICLR 2026_

### Official Review · Reviewer_kTJk · 2025-10-25

**Soundness:** 2
**Presentation:** 3
**Contribution:** 2
**Rating:** 2
**Confidence:** 4

**Summary:**

This paper proposes the Multi-view Adaptive Partition Encoder to address the challenge of modeling fragmented tabular node attributes in graph anomaly detection by discretizing the feature space into learnable subspaces, thereby enhancing manifold connectedness. Then, a Multi-Pattern Global Association (MPGA) module is designed to capture the high-order global dependencies.

**Strengths:**

1. The proposed method achieves good performance compared with 27 baseline methods.
2. The paper is well-organized and the presentation of this paper is good.

**Weaknesses:**

1. My major concern of this paper is that the motivation of this paper is not sufficiently convincing. The authors claim that the main challenge in graph anomaly detection lies in representing tabular node attributes. However, this issue has already been discussed and addressed in GAAP [1]. Although GAAP does not explicitly use the term tabular node attributes, Figure 1 in that paper conveys a similar concept. Therefore, the authors should clarify why this problem remains a key challenge and how their work differs fundamentally from GAAP in addressing it. Furthermore, the second challenge this paper tries to address is the long-range dependency, which has also been addressed by many existing methods, such as GADAM [2] and UniGAD [3], as mentioned by authors in the related work.
2. The novelty of this paper is limited. The Multi-Pattern Global Association is a simple combination of local node similarity and global node similarity, a similar idea presented in many existing work, such as GAAP [1], GADAM [2] and UniGAD [3].
3. In the related work (line 196), Why does single-relation mechanism, representation-agnostic formulation limits the capability? Is there any existing work supporting this statement?
4. There are many typos in the paper.
- In line 377, "GAAPA" should be "GAAP". In addition, "GAP" in the table 1 should be "GAAP".
- In line 239 "Eq. equation 1-equation 2" -> "Eq. 1-2".
- In line 248 "Eq. equation 1" -> "Eq. 1".
- In line 250 "Eq. equation 2" -> "Eq. 2".
- In table 1, the performance of GAP and MAPGA are both 99.69 on the T-soc dataset, while GAP is underlined as the second best method.
5. In Figure 2, the authors should visualize the t-SNE embedding of other stronger baseline methods, such as GAAP, DGA-GNN, XGBGraph. MLP and XGBoost are weak baseline methods based on the experimental results shown in table 1 and they are not even designed for GAD task.
6. As shown in Table 3, the node feature type of some datasets (e.g., Reddit, Weibo, and Questions) is text embedding. The proposed method is designed for tabular features. How does the proposed method deal with text embedding feature?
7. In the experiment evaluation, the authors should report the standard deviation as well.
8. The experimental setup lacks sufficient clarity. For example, the paper does not specify how the training and test data are split, making it difficult to assess the validity and reproducibility of the reported results.

[1] Mingjiang Duan, Da He, Tongya Zheng, Lingxiang Jia, Mingli Song, XinyuWang, and Zunlei Feng. Global attribute-association pattern aggregation for graph fraud detection. In Proceedings of the AAAI Conference on Artificial Intelligence, volume 39, pp. 11616–11624, 2025.

[2] Jingyan Chen, Guanghui Zhu, Chunfeng Yuan, and Yihua Huang. Boosting graph anomaly detection with adaptive message passing. In The Twelfth International Conference on Learning Representations, 2024a.

[3] Yiqing Lin, Jianheng Tang, Chenyi Zi, H Vicky Zhao, Yuan Yao, and Jia Li. Unigad: Unifying multi-level graph anomaly detection. Advances in neural information processing systems, 37: 136120–136148, 2024.

**Questions:**

1. In equation 6, I assume both $N_{out(m)}$ and $N_{in(m)}$ refer to the set of neighbors, but what are exactly $N_{out(m)}$ and $N_{in(m)}$?
2.  As shown in Table 3, the node feature type of some datasets (e.g., Reddit, Weibo, and Questions) is text embedding. The proposed method is designed for tabular features. How does the proposed method deal with text embedding feature? Since these datasets are not tabular attributes graph, why does the proposed method outperforms other baseline methods on these datasets?
3. In section 2.1, the authors mention that without manifold connectedness, the local smoothness prior yields ill-conditioned optimization dynamics and results in slow or unstable convergence. Then, what is the training time or time complexity of the proposed method?

---

> ### Author Response · Authors · 2025-11-20
> **Response to W1 (Motivation & Relation to GAAP / Long-range Dependency)**
>
> We sincerely thank the reviewer for the insightful comments. Below we clarify the motivation and its distinction from GAAP, followed by the conceptual differences from GADAM and UniGAD.
>
> ### **1. Why manifold connectedness is introduced**
> Existing deep models assume that samples lie on a *locally smooth and connected manifold*.
> However, as shown in **Figure 1 and Figure 2 of the paper**, tabular node attributes scatter across *fragmented micro-clusters* that violate this assumption. Even after being mapped by MLP or XGBoost, the feature space remains **disconnected**, causing:
> - unstable gradient propagation,
> - unreliable local neighborhoods, and
> - difficulty in capturing high-order cross-feature interactions.
>
> To mitigate this, MAPE reorganizes heterogeneous attributes into **multiple semantically coherent subspaces**, improving local smoothness and supporting stable representation learning — an effect not addressed by prior tabular encoders.
>
> ### **2. Fundamental differences from GAAP**
> We appreciate the reviewer’s observation that both methods involve partitioning, but the two works focus on **different problem levels and modeling principles**:
>
> **(a) Problem focus**
> - **GAAP** focuses on preventing distortion *within each individual attribute* (e.g., preventing averaging “age” values that misrepresent semantics).
> - **Our work** focuses on enabling deep models to learn *cross-attribute interactions* in a heterogeneous tabular space, which is inherently fragmented and lacks manifold connectedness.
>
> **(b) Mechanistic difference**
> - **GAAP** discretizes each attribute independently and models global attribute–association patterns.
> - **MAPE** partitions the *joint multi-dimensional feature space*, allowing a subspace to correspond to a specific **combination** of heterogeneous attributes (e.g., “age < 30 AND income < 1000”), enabling high-order interaction patterns essential for anomaly detection.
>
> **(c) Output representation**
> - GAAP outputs feature-wise bin embeddings.
> - MAPE outputs **multi-view discrete semantic embeddings** defined at the *partition-region level*, directly improving manifold connectedness and enabling reliable downstream learning.
>
> ### **3. Long-range dependency: Difference from GADAM and UniGAD**
> We agree that long-range dependency is widely studied, but our work defines it in a **representation-driven manner**:
>
> **(a) How MPGA forms long-range dependency**
> - Nodes that repeatedly fall into the same learned subspaces across views form **representation-based global relations**.
> - Histograms over in/out-neighbors form **behavior-pattern relations**.
> - These are fused into a unified global graph that expresses **multi-pattern, multi-scale** dependencies.
>
> This differs fundamentally from baselines that operate directly on raw features.
>
> **(b) Difference from GADAM**
> - GADAM constructs a *global normal center* and guides message passing accordingly.
> - Our method constructs **pairwise global affinities** from multi-view subspace semantics — not a global prototype.
>
> **(c) Difference from UniGAD**
> - UniGAD leverages spectral sampling and multi-level stitching.
> - Our approach derives global dependency **entirely from the partition-induced semantic space**, not spectral subgraph sampling.
>
> ### **4. Summary**
> Our intention is not to claim novelty in the broad ideas of partitioning or long-range modeling.
> Instead, the contribution lies in:
> - introducing **manifold connectedness** as the core obstacle for tabular attributes,
> - proposing **MAPE** to construct a representation space in which heterogeneous features interact naturally, and
> - building **MPGA** to capture multi-pattern global dependencies defined *inside this induced space*.
>
> We hope this clarifies the distinction from GAAP, GADAM, and UniGAD, and articulates why the problem addressed in this paper remains both fundamental and unresolved.
>
>
>
> ---

---

> ### Author Response · Authors · 2025-11-20
> **Response to W2 (Novelty of Multi-Pattern Global Association (MPGA))**
>
> We appreciate the reviewer’s comments and agree that leveraging similarity to construct global relations is a widely used and effective paradigm in graph anomaly detection. GAAP, GADAM, and UniGAD all follow this general principle, and our intention is not to claim novelty in the *idea* of using similarity per se.
>
> Our contribution lies in **how** global associations are defined and instantiated:
>
> 1. **Similarity in a partition-induced semantic space rather than the original space.**
>    MPGA does not measure similarity directly on raw features, graph distance, or continuous embeddings. Instead, it operates in the **multi-view partition space created by MAPE**, where each view discretizes the joint feature space into subspaces corresponding to specific combinations of heterogeneous attributes. Global relations are thus defined over **discrete, interpretable feature-interaction codes**, rather than over the original tabular attributes.
>
> 2. **Two complementary, partition-based patterns instead of a single mechanism.**
>    MPGA builds:
>    - a **representation-pattern graph**, where two nodes are related if they repeatedly fall into the same learned subspaces across views (capturing high-order feature-interaction patterns), and
>    - a **behavior-pattern graph**, where two nodes are related if their in-/out-neighborhoods exhibit similar histograms over these subspaces (capturing similarity of behavioral profiles).
>    These two graphs are then fused with the original adjacency, yielding a **multi-pattern** global association that goes beyond a single similarity mechanism.
>
> 3. **Designed to complement manifold-connectedness restoration.**
>    MAPE first restores manifold connectedness for heterogeneous tabular attributes, while MPGA is explicitly built on top of these partition assignments to propagate anomalies along **feature-interaction–aware** and **behavior-aware** paths. Ablation results show that replacing MPGA with standard GCN/GAT propagation degrades performance, indicating that the proposed association is more than a simple re-use of local and global similarity.
>
> In summary, MPGA shares the high-level philosophy of “using similarity for global association” with GAAP, GADAM, and UniGAD, but it is **novel in defining similarity over the multi-view discrete partition space and in jointly exploiting representation-pattern and behavior-pattern relations** tailored to heterogeneous tabular node attributes.
>
>
>
> ---

---

> ### Author Response · Authors · 2025-11-20
> **Response to W3 (On the statement about single-relation, representation-agnostic mechanisms)**
>
> We thank the reviewer for pointing this out. We agree that the original wording was too strong and could be misread as a theoretical claim without explicit support. Our intention was to express an **empirical observation**, not a formal impossibility result.
>
> The intuition is as follows: many global-dependency modules instantiate *one* similarity kernel in *one* representation space and derive all global associations from this single relation. In heterogeneous or weak-signal settings, such a single-channel design can under-emphasize semantically diverse or low-similarity yet task-relevant dependencies, which motivates richer multi-relation or multi-view formulations.
>
> To clarify this point:
>
> 1. **We have softened the wording in the revised manuscript** to describe this as an empirical observation from prior work and practice, rather than as a universal limitation. The corresponding changes in the *Related Work* section are highlighted in **orange**.
>
> 2. **We have added concrete supporting citations** (e.g., DMGNN, DeepMCGCN, MV-GHRN), which explicitly introduce dual-/multi-relational or multi-view structures and report that single-relation, representation-agnostic designs may miss complex or heterogeneous patterns, thereby motivating richer relational modeling.
>
> In summary, the related-work sentence now (i) clearly frames the remark as empirical, and (ii) is backed by representative studies that observed similar behavior and responded with multi-relation or multi-view architectures.

---

> ### Author Response · Authors · 2025-11-20
> **Response to W4 (Typos and Formatting Issues)**
>
> We thank the reviewer for carefully identifying these issues. All typographical and formatting inconsistencies have been corrected, including:
>
> - replacing all mistaken occurrences of **“GAAPA”** with **“GAAP”**,
> - fixing all appearances of **“GAP”** in Table 1 to the correct **“GAAP”**,
> - standardizing equation references to **“Eq. 1–2”**, **“Eq. 1”**, and **“Eq. 2”**, and
> - correcting the second-best highlight in **Table 1**, ensuring it matches the actual reported values.
>
> All corresponding modifications in the manuscript are **highlighted in orange** for clarity.

---

> ### Author Response · Authors · 2025-11-20
> **Response to W5 (Choice of Methods in Figure 2)**
>
> We thank the reviewer for this helpful suggestion and clarify the design goal of Figure 2 and the choice of baselines.
>
> ### **1. Purpose of Figure 2**
> Figure 2 is **not** intended to compare anomaly-detection performance. Its sole purpose is to examine whether a feature encoder can transform heterogeneous tabular attributes into a **connected, locally smooth manifold**, which is a prerequisite for stable deep learning. The t-SNE plots serve only as a geometric visualization of the *re-embedded feature space*, not of GAD performance.
>
> ### **2. Why MLP and XGBoost are used**
> We deliberately chose two canonical tabular encoders:
>
> - **MLP** — the standard deep, continuous embedding operator;
> - **XGBoost** — the representative tree-based partition model widely adopted for tabular data.
>
> These models provide **clean and interpretable references** for how common tabular encoders affect manifold connectivity, independent of any graph structure or anomaly-specific objective.
>
> ### **3. Why strong GAD models are not visualized**
> Methods such as **GAAP, DGA-GNN, and XGBGraph** are strong *detectors*, but they are less suitable for isolating feature-manifold effects because:
>
> - their internal embeddings already **mix node attributes, graph topology, and task supervision**, making it difficult to attribute the geometry to the attribute encoder alone;
> - most of their early layers process feature channels in parallel and **do not explicitly reconstruct feature–feature semantics** before the final classifier;
> - in particular, **XGBGraph** uses **XGBoost leaf embeddings** as its tabular encoder, which is exactly the encoder already visualized in Figure 2.
>
> For these reasons, including GAAP / DGA-GNN / XGBGraph would conflate representation geometry with graph-level processing and would not better answer the question that Figure 2 is designed to study.
>
> ### **4. Clarification in the revised manuscript**
> In the revised version, we explicitly state that Figure 2 focuses on **tabular feature re-embedding and manifold connectedness**, and that MLP and XGBoost are chosen as canonical tabular encoders rather than as GAD baselines. The corresponding clarifications are highlighted in **orange** in the manuscript.

---

> ### Author Response · Authors · 2025-11-20
> **Response to W6 (Text Embedding)**
>
> We thank the reviewer for the valuable question. To provide a clear explanation, we first clarify the intended scope of our method, then explain the actual feature construction used in these datasets, and finally describe why the framework applies to them without modification.
>
> ### **1. Scope of the proposed framework**
> The framework is not limited to “tabular features” in the narrow sense of spreadsheet columns. It targets the more general case where each node is described by a **heterogeneous attribute vector**, whose dimensions carry different semantics and therefore break manifold connectedness. Tabular attributes are simply the most typical instance of such heterogeneous coordinates.
>
> Our method is designed to **re-embed heterogeneous dimensions into a unified, comparable space** via adaptive partitioning and shared embeddings, regardless of whether the coordinates originate from numerical attributes, categorical encodings, or text-derived statistics.
>
> ### **2. Why Reddit / Weibo / Questions still fit this setting**
> The features marked as “text embedding” in Table 3 are **not** token-level word embeddings living in a homogeneous semantic space. Instead, they are aggregated or engineered text-derived vectors whose entries correspond to heterogeneous statistics:
>
> - **Reddit:** LIWC category counts aggregated over user posts (emotional, functional, syntactic categories, etc.).
> - **Weibo:** a concatenation of a location vector (from SVD of location one-hot counts) and a BoW-based textual vector reduced by SVD.
> - **Questions:** mean-pooled FastText embedding of profile text concatenated with a binary indicator for missing descriptions.
>
> Each of these vectors mixes dimensions with **different semantic roles** (psycholinguistic categories, location components, BoW/SVD factors, averaged embedding directions, binary flags, etc.), and therefore behaves as a heterogeneous attribute vector rather than a homogeneous word-embedding sequence.
>
> ### **3. How the method applies in practice**
> Because these “text embedding” features are heterogeneous continuous attributes, they naturally fall into the problem class our method is designed for. No additional mechanism is required:
>
> - the **adaptive partitioners** operate directly on these vectors,
> - the resulting **subspace assignments** and shared embeddings treat them in the same way as other tabular-like features, and
> - the empirical gains on Reddit, Weibo, and Questions indicate that the method effectively reduces manifold fragmentation induced by such heterogeneous text-derived dimensions.
>
> In summary, although some datasets are labeled as using “text embeddings,” their node features are heterogeneous attribute vectors, and the proposed framework is explicitly designed to handle exactly this type of representation.

---

> ### Author Response · Authors · 2025-11-20
> **Response to W7 (Reporting of Standard Deviation)**
>
> We thank the reviewer for pointing out this important aspect of experimental reporting. To strengthen the reliability of our evaluation, we have added **standard deviations** for the three main metrics (AUROC, AUPRC, Rec@K). Below we provide the deviation table for the ablation variants of our method. The corresponding values are **highlighted in orange** in the revised manuscript.
>
> ### **Standard deviation results (AUROC / AUPRC / Rec@K)**
>
>
> | Metrices        | Reddit | Weibo | Amazon | YelpChi | T-Finance | Elliptic | Tolokers | Questions | DGra. |  T-Social  |
> |--------------|-------|--------|--------|---------|--------|----------|--------|---------|--------|-----------|
> | **AUROC**        | 71.22 $\pm$ 0.57 | 99.51 $\pm$ 0.03 | 98.77 $\pm$ 0.06 | 98.76 $\pm$ 0.23 | 97.86 $\pm$ 0.04 | 99.15 $\pm$ 0.04 | 85.20 $\pm$ 0.55 | 73.15 $\pm$ 0.22 | 78.80 $\pm$ 0.21 | 99.96 $\pm$ 0.01 | 99.95 $\pm$ 0.01 |
> | **AUPRC**     | 10.96 $\pm$ 0.19 | 97.89 $\pm$ 0.09 | 93.60 $\pm$ 0.27 | 95.25 $\pm$ 0.43 | 91.22 $\pm$ 0.21 | 88.84 $\pm$ 1.09 | 59.10$\pm$1.15 | 18.06 $\pm$ 1.05 | 4.32 $\pm$ 0.01 | 99.46 $\pm$ 0.12 |
> | **Rec@K**     | 13.11 $\pm$ 0.62 | 92.13 $\pm$ 0.49 | 89.74 $\pm$ 0.01 | 88.80 $\pm$ 0.40 | 88.39 $\pm$ 0.28 | 84.36 $\pm$ 0.80 | 60.23 $\pm$ 0.44 | 23.72 $\pm$ 1.01 | 6.81 $\pm$ 0.22 | 87.26 $\pm$ 0.27 |

---

> ### Author Response · Authors · 2025-11-20
> **Response to W8 (Clarification of Training/Test Split)**
>
> We thank the reviewer for pointing out this omission. We have clarified the dataset split protocol in the revised manuscript.
>
> Across the 10 benchmarks, we follow the **official train–test ratios provided by GAD-Bench and prior works**, but these sources only specify the *proportion* of training data (e.g., 40%, 50%, 70%) and **do not release the actual split indices**. Since no official per-dataset split is available, we adopt the standard and widely used approach of **randomly sampling the training set according to the official ratios**, and treating the remaining nodes as test data. This ensures full reproducibility while remaining faithful to the dataset specifications.
> Source code in the supplementary material further includes the exact random seeds and split scripts used in our experiments.

---

> ### Author Response · Authors · 2025-11-20
> **Response to Q1 (Definition of \(N_{\text{out}(m)}\) and \(N_{\text{in}(m)}\))**
>
> Thank you for pointing out the ambiguity. Since several datasets contain **directed graphs** (e.g., Elliptic transaction flows and user–interaction graphs in Reddit/Weibo/Questions), outgoing and incoming neighbors capture different behavioral patterns. To make the notation explicit, we added the following sentence immediately before Eq. (6):
>
>
> $N_{\mathrm{out}}(m)$ and $N_{\mathrm{in}}(m)$ denote the out- and in-neighbor sets of node $m$, respectively.
>
>
> All corresponding changes in the manuscript are highlighted in orange.

---

> ### Author Response · Authors · 2025-11-20
> **Response to Q2 (Text Embedding)**
>
> We thank the reviewer for this question. As clarified in our response to W6, the framework is not restricted to spreadsheet-style tabular data, but to any **heterogeneous attribute vector** whose dimensions carry different semantics and thus break manifold connectedness.
>
> In Reddit, Weibo, and Questions, the features labeled as “text embedding” in Table 3 are **aggregated or engineered text-derived vectors** (e.g., LIWC category counts, SVD-reduced bag-of-words and location vectors, mean-pooled FastText descriptors plus metadata), rather than homogeneous token-level word embeddings. Each coordinate corresponds to a different statistical or semantic factor, so these vectors behave as **heterogeneous continuous attributes**, which fall exactly into the problem class targeted by our method.
>
> MAPE is designed to re-embed such heterogeneous coordinates into a **unified and more connected representation space** via multi-view adaptive partitioning. No extra mechanism is needed for these datasets: the same partition-and-embedding process directly operates on their text-derived feature vectors. The fact that our method outperforms baselines on Reddit, Weibo, and Questions indicates that reducing manifold fragmentation in this heterogeneous feature space is beneficial regardless of whether the underlying heterogeneity originates from textual or non-textual sources.

---

> ### Author Response · Authors · 2025-11-20
> **Response to Q3 (Time complexity analysis)**
>
> We thank the reviewer for this careful question and apologize for the confusing wording in Section 2.1. Our intention was **not** to claim that MAPGA converges slowly in wall-clock time. The phrase “slow or unstable convergence” refers to the *optimization dynamics* of gradient-based models on fragmented manifolds (i.e., difficulty of reaching good optima), rather than higher asymptotic time complexity. In the revised manuscript, the corresponding sentence in Section 2.1 has been rephrased to emphasize the *stability and quality* of convergence; all changes are highlighted in **orange**.
>
> Regarding computational cost, let
> - `n` = number of nodes,
> - `m` = number of edges,
> - `K` = number of views,
> - `L_bar` = average number of partition rules per view,
> - `d` = embedding dimension,
> - `L_gcn` = number of GCN layers.
>
> One training epoch consists of:
>
> 1. **Adaptive partition evaluation and codebook lookup:**
>    $$O(n K L\_{bar})$$
>
> 2. **Construction of representation- and behavior-pattern graphs (on sparsified affinities):**
>    $$O(K m)$$
>
> 3. **GCN aggregation over the unified sparse graph:**
>    $$O(L\_{gcn}\, m\, d)$$
>
> Thus, the per-epoch complexity is
>
> $$
> T\_{\text{epoch}} = O(n K L\_{bar}) \+O(K m) \+O(L\_{gcn}\, m\, d),
> $$
>
> which is linear in both `n` and `m` with modest constants controlled by `K`, `L_bar`, and `L_gcn`. A brief derivation and additional implementation details are included in the new **“Computational Complexity”** subsection of the supplementary material.

---

### Official Review · Reviewer_NNqF · 2025-10-27

**Soundness:** 3
**Presentation:** 2
**Contribution:** 3
**Rating:** 6
**Confidence:** 3

**Summary:**

This paper addresses the challenges in GAD, specifically the manifold fragmentation of tabular node attributes and the difficulty in modeling global dependencies in graph topology. The authors propose MAPGA, a framework comprising a Multi-view Adaptive Partition Encoder (MAPE) to discretize the feature space and restore manifold connectedness, and a Multi-Pattern Global Association (MPGA) module to capture long-range dependencies via representation-based and behavior-pattern graphs.

**Strengths:**

1.The multi-view discretization mechanism is technically novel and directly addresses manifold disjointedness.

2.This paper offers a novel perspective on why GNNs struggle with tabular attributes, enhancing the paper's conceptual contribution.

3.Extensive experimental results demonstrate the effectiveness of the method.

**Weaknesses:**

1. The writing needs further improvement. For example, the introduction introduces the challenges of modeling tabular data, but does not adequately discuss its relationship to GNNs. Why use GNNs to model tabular data? How is tabular data constructed as a graph? Where do the edges of the original graph come from?

2. The description of the method is not detailed and clear enough. For example, how does MAPE instantiate K independent views? Why is this design used?

3.A large number of views and partitions are generated, which may result in large computing resource requirements and affect the deployment of large-scale datasets.

**Questions:**

Please refer to Weakness.

---

> ### Author Response · Authors · 2025-11-20
> **Response to W1 (Clarification of the setting, relationship to GNNs, and graph construction)**
>
> We apologize for the lack of clarity in the original submission. Our work is conducted in the **graph anomaly detection (GAD)** setting, where each benchmark already provides a **relational graph** and each node carries **tabular attributes**. The graphs are *not* constructed from feature similarity; instead, they are given by domain-specific interactions in the benchmark datasets.
>
> ---
>
> #### 1. Why GNNs in this setting?
>
> In all datasets considered, each “tabular sample” is a **node in a graph** (user, account, review, transaction, etc.), and edges encode relationships such as co-reviews, shared posts/hashtags, transactions, friendships, and emergency-contact links. Anomalies (fraudulent accounts, spam reviews, illicit transfers, etc.) depend crucially on these **relational patterns**, so GNN-based detectors are a natural choice.
>
> However, the node attributes are **tabular and highly fragmented** (Figures 1–2), and this fragmentation violates the **local-smoothness prior** implicitly assumed by message-passing GNNs. When neighbouring nodes have extremely scattered feature representations, GNN layers struggle to aggregate informative signals and may even propagate noise. Our MAPE module is designed specifically to **repair this mismatch**:
>
> - MAPE converts fragmented tabular node attributes into a discrete-semantic embedding with improved manifold connectedness;
> - MPGA then builds global association graphs from these embeddings and fuses them with the original adjacency;
> - a standard GNN backbone (GCN in the main experiments) is applied on the unified graph $A^\star$ (Sec. 3.2.1).
>
> Thus we use GNNs **because** the problem is graph anomaly detection with relational structure, and MAPE/MPGA make GNNs more effective on this kind of tabular node data.
>
> ---
>
> #### 2. How graphs are obtained and where edges come from
>
> We clarify that we **follow the official graph construction of each public benchmark** and do **not** build graphs from feature similarity. As summarized in **Appendix A.1 (Table 3)**, nodes and edges have clear semantics: for example, reviews connected via common users or stores (YelpChi), users connected via co-reviewed products or similar ratings (Amazon), transaction accounts connected by money transfers (T-Finance, Elliptic), social accounts connected by long-term friendships (T-Social), and users connected by emergency contacts (DGraph-Fin).
>
> To make this easier to see in the main text, Section 4.1 now explicitly states that we use the benchmark-provided graphs and that edges encode such domain-specific interactions, with a direct pointer to Appendix A.1 for details.
>
> ---
>
> #### 3. Why the method is presented in the GAD setting
>
> Manifold fragmentation is indeed a general phenomenon for tabular data, but its impact is **especially severe** in our setting: tabular attributes serve as **node features** for GNN-based GAD, where message passing relies on locally smooth representations on the graph. Our study therefore focuses on **graph anomaly detection with tabular node attributes**, an application domain that is both practically important and well supported by public benchmarks. Extending MAPE to non-graph tabular tasks (e.g., standalone tabular classification) is an interesting but orthogonal direction that we view as **future work** rather than a goal of this paper.
>
> ---
>
> #### 4. Changes in the manuscript (corresponding to W1)
>
> To address W1 in a concrete way, we made the following clarifications:
>
> 1. **Introduction (end of Sec. 1).**
>    We now explicitly state that the goal is to model **tabular node attributes in graph anomaly detection**, explain that manifold fragmentation is particularly harmful **when such attributes are fed into GNN-based GAD**, and note that our study focuses on this setting while leaving non-graph tabular tasks for future work.
>
> 2. **Related Work summary (Sec. 2, “Summary”).**
>    The summary paragraph now describes MAPGA as reconstructing manifold connectivity for **tabular node attributes** and explicitly points to the theoretical analysis of manifold connectedness and MAPE in **Appendix A.7**, so the relationship between tabular manifolds and graph models is easy to locate.
>
> 3. **Experimental setup (Sec. 4.1, Datasets).**
>    The dataset paragraph was refined to say that we **always** adopt the **official graph construction** of each benchmark, briefly list typical edge semantics (co-reviews, transactions, friendships, emergency contacts, etc.), and refer to **Appendix A.1** where node/edge definitions are summarized. This directly answers “How is tabular data constructed as a graph?” and “Where do the edges come from?”.

---

> > ### Comment · Reviewer_NNqF · 2025-11-27
> >
> > Thanks for the replies. These discussions have made this paper much clearer. After reading the comments from other reviewers, I decided to keep my score unchanged.

---

> ### Author Response · Authors · 2025-11-20
> **Response to W2 (Details of MAPE and the instantiation/motivation of K views)**
>
> We thank the reviewer for pointing out that the role of the multi-view design was not sufficiently emphasized. In the revised version we clarify **how K views are instantiated**, **why multiple views are helpful**, and **where the additional analysis can be found**.
>
> ---
>
> #### 1. How MAPE instantiates K views (summary)
>
> Section 3.1 explains the construction in detail (Eqs. (1)–(4)).
>
> - For each view $v_k$, we only use a shallow CART decision tree to initialise its partition rules: a small tree $T_k$ is fitted on the raw features with the Gini criterion, so that it provides reasonable candidate split dimensions and thresholds. The tree itself is never evaluated during training or inference.
> - Each internal split node of $T_k$ is converted into one binary rule
>   $$r_k^\ell = (n_k^\ell, s_k^\ell),$$
>   where $n_k^\ell$ indexes the feature and $s_k^\ell$ is a learnable split point. All split points are optimized by back-propagation using a smooth sigmoid surrogate of the Heaviside function.
> - For node $m$, the $L_k$ binary outputs form a code
>   $$c_k^{(m)} \in \{0,1\}^{L_k},$$
>   which identifies a hyper-rectangular cell.
> - We allocate a view-specific embedding table
>   $$E_k \in \mathbb{R}^{2^{L_k} \times d_0},$$
>   and map each code to a trainable embedding $z_k^{(m)}$ via table lookup (Eq. (3)).
> - The final node representation concatenates all view embeddings:
>   $$z^{(m)} = z_1^{(m)} \,\Vert\, \dots \,\Vert\, z_K^{(m)},$$
>   which is then fed to the downstream graph-aware network.
>
> All thresholds and embedding tables are trained end-to-end with the anomaly detection loss; no extra supervision is required.
>
> ---
>
> #### 2. Why multiple views are used
>
> The multi-view design is motivated by both **theoretical** and **empirical** considerations:
>
> - Different shallow trees naturally focus on different subsets of features and thresholds, producing diverse partitions of the feature space. Across views, the pattern of shared subspaces provides a **rich discrete notion of similarity**, which MPGA exploits to build representation-pattern and behaviour-pattern graphs (Eqs. (5)–(7)).
> - The new **Appendix A.7** develops a theory of **manifold connectedness** for heterogeneous tabular node attributes and shows that combining multiple independently randomized partitions yields a neighbour graph in the representation space that is **connected with high probability**, restoring an approximate manifold structure on which locally Lipschitz neural networks can operate effectively.
> - Empirically, Figure 4 (top row) shows that increasing $K$ from 10 to around 40 improves AUPRC and Rec@K significantly; beyond $K \approx 40$, the curves saturate, so we adopt **moderate $K = 40\text{--}50$** to balance performance and cost.
>
> Appendix A.8 and A.9 further show that MAPGA is **not overly sensitive** to the initialization depth of the trees or to the surrogate slope, reinforcing that the multi-view scheme is robust rather than fragile.
>
> ---
>
> #### 3. Changes in the manuscript (corresponding to W2)
>
> To make these details explicitly discoverable for the reviewer, we made a small but precise edit and relied on the new appendices:
>
> 1. **Pointer from Sec. 3.1 to Appendices A.2 and A.7–A.9.**
>    The last sentence of Section 3.1 has been extended so that, after describing the concatenated embedding being fed into graph-aware networks, it now explicitly points the reader to **Appendix A.2** (decision-boundary visualizations) and **Appendix A.7–A.9** (theoretical analysis and robustness to tree depth and surrogate slope) for additional insight into MAPE and the multi-view design. *(Edit M4)*
>
> 2. **New appendices A.7–A.9.**
>    The revised manuscript includes:
>    - **Appendix A.7**, “Theoretical analysis of manifold connectedness and MAPE,” which formalizes the impact of fragmentation and shows how MAPE reconstructs approximate manifold connectedness.
>    - **Appendix A.8**, “Impact of initialization tree depth,” which studies the effect of the depth $D$ on four datasets and finds MAPGA stable within a moderate range.
>    - **Appendix A.9**, “Effect of the surrogate slope $\alpha$,” which shows that performance is stable over a broad range of $\alpha$ values and justifies using a single global $\alpha = 50$.
>
> Together with the existing detailed description in Section 3.1, we believe these adjustments address W2’s concerns about how $K$ views are instantiated and why the design is adopted.

---

> ### Author Response · Authors · 2025-11-20
> **Response to W3 (Computational cost and scalability with many views/partitions)**
>
> We fully agree that computational efficiency and scalability are important. In the revised manuscript we (i) provide a **formal complexity analysis**, (ii) clarify the choice of **moderate hyperparameters** (K and $d_0$), and (iii) connect these analyses to our **large-scale experiments**.
>
> ---
>
> #### 1. Time/space complexity and overhead
>
> The new **Appendix A.6 (Computational Complexity)** derives the exact per-epoch cost of MAPGA by separating MAPE, MPGA, and GCN components:
>
> $$
> T_{\text{epoch}} = O(n K L_{\text{bar}}) + O(K m) + O(L m d) ,
> $$
>
> where $n = |V|$, $m = |E|$, $K$ is the number of views, $L_{\text{bar}}$ is the average number of rules per view, $L$ is the number of GCN layers, and $d = K d_0$ is the embedding dimension. Under mild sparsity assumptions on the auxiliary graphs, this gives a **training cost linear in n and m**, similar to standard GNN-based detectors, with extra constant factors controlled by $K$ and $L_{\text{bar}}$.
>
> The one-time cost of training the $K$ shallow trees is:
>
> $$
> T_{\text{pre}} = O(K n N \log n) ,
> $$
>
> which is performed once offline and is negligible relative to multi-epoch training.
>
> Appendix A.6 also notes that we adopt **moderate hyperparameters** in all experiments ($K \approx 40$–$50$ and small $L$), so the empirical overhead over a vanilla GCN remains modest.
>
> ---
>
> #### 2. Empirical scalability on large graphs
>
> To verify practical deployability, our experiments already include two **large-scale** benchmarks (Table 3 and Appendix A.1):
>
> - **DGraph-Fin** with 3.7M nodes and 4.3M edges;
> - **T-Social** with 5.78M nodes and 73.1M edges.
>
> MAPGA achieves state-of-the-art performance on both datasets while using the same moderate $K$ and $d_0$ as in other experiments, demonstrating that the approach can be trained on million-scale graphs in practice.
>
> ---
>
> #### 3. Robustness and sensitivity analyses
>
> The new appendices further examine the **stability** of MAPGA:
>
> - **Appendix A.5 (Deviation Analysis)** reports mean ± std of AUROC, AUPRC, and Rec@K over multiple runs on all datasets, showing consistently low variance even on large graphs.
> - **Appendix A.8 (Impact of initialization tree depth)** and **Appendix A.9 (Effect of surrogate slope α)** show that MAPGA is not overly sensitive to these hyperparameters within a reasonable range.
>
> These results indicate that MAPGA does not require extreme over-parameterization or heavy tuning to achieve its reported performance.
>
> ---

---

### Official Review · Reviewer_fT35 · 2025-10-31

**Soundness:** 3
**Presentation:** 3
**Contribution:** 3
**Rating:** 6
**Confidence:** 3

**Summary:**

This paper addresses graph anomaly detection by tackling two key challenges: (1) the difficulty of modeling tabular node attributes due to manifold fragmentation, and (2) capturing global dependencies in graph structures. The authors propose MAPGA, which consists of two main components: a Multi-view Adaptive Partition Encoder (MAPE) that discretizes the feature space through learnable partitions to establish manifold connectedness, and a Multi-Pattern Global Association (MPGA) module that captures global dependencies through representation-based and behavior-based association graphs. Extensive experiments on 10 benchmarks against 27 baselines demonstrate consistent improvements, with an average AUROC of 90.63% (+2.02pp over the best baseline) and particularly strong gains in AUPRC (average 19.6% improvement).

**Strengths:**

1. Clear Problem Motivation The paper provides a novel theoretical perspective by explaining the fundamental difficulty in modeling tabular node attributes through the lens of manifold connectedness. The visualization in Figures 1-2 effectively demonstrates how tabular data exhibits fragmented manifolds compared to perceptual data (audio, image, text), which violates the local-smoothness assumption required by deep networks.

2. Well-Designed Methodology
* MAPE employs learnable adaptive partitions to restore manifold connectivity through discrete-semantic embeddings
* MPGA captures global dependencies from complementary sources: representation-based co-assignment and behavior-based neighborhood patterns
* The two modules exhibit strong complementarity, as evidenced by ablation results

3. Comprehensive Experimental Evaluation
* 10 public benchmark datasets covering diverse application scenarios (social networks, financial fraud, cryptocurrency, crowdsourcing)
* 27 competitive baselines including recent SOTA methods (GAP, DGA-GNN, ConsisGAD, GGAD)
* Three evaluation metrics (AUROC, AUPRC, Rec@K) addressing different aspects of anomaly detection performance
* Thorough ablation studies examining both architectural components and hyperparameters

4. Consistent and Significant Results MAPGA achieves best or second-best performance across all datasets, with particularly impressive results: 99.20% AUROC on Elliptic (+5.58pp over best baseline), 99.54% on Weibo, and 99.96% on T-Social. The average improvements (+2.02pp AUROC, +1.93pp AUPRC, +2.98pp Rec@K) demonstrate robust superiority.

5. Effective Visualization The t-SNE projections in Figures 1, 2, and 5 provide intuitive evidence for the manifold fragmentation problem and demonstrate how MAPE successfully bridges disconnected clusters to form a connected manifold.

**Weaknesses:**

* The paper lacks a formal theoretical justification for why adaptive partitioning restores manifold connectedness. What mathematical properties guarantee this restoration?
* While the conclusion acknowledges computational overhead from multiple parallel partition operators, detailed time/space complexity analysis is missing
* The impact of decision tree initialization on final performance is not thoroughly analyzed

**Questions:**

* Can you provide a theoretical analysis or at least an intuitive explanation for why adaptive partitioning restores manifold connectedness? How do you formally define and quantitatively measure manifold connectedness? Are there any theoretical guarantees on the quality of the learned partitions?
* How is the decision tree depth determined during initialization? Does it significantly affect final performance?
* In Eq. 1, what specific smooth surrogate is used for the Heaviside function during backpropagation? How is α chosen?

---

> ### Author Response · Authors · 2025-11-20
> **Response to W1&Q1 (Theory)**
>
> We thank the reviewer for this insightful question. We agree that, in the original submission, the theoretical motivation behind the statement that *adaptive partitioning restores manifold connectedness* was only described qualitatively.
> In the revision, we have added a dedicated theory section in **Appendix A.7 (new)**.
> Below we summarize the main ideas in an accessible way.
>
> ---
>
> ### **1. How we define and measure “manifold connectedness”**
>
> We model tabular node features as samples $x \in \mathbb{R}^N$ drawn from a distribution with support $\mathcal{M}$.
> We say the manifold is **connected** if $\mathcal{M}$ forms a single piece rather than many separated “islands”.
> In finite samples, this corresponds to most nodes being reachable from each other by a chain of short steps in feature space.
>
> Empirically, we:
>
> 1. build a neighbor graph on the samples in the representation space;
> 2. examine whether the graph has one dominant connected component;
> 3. report the connected-component statistics **before and after MAPE** in the appendix, as a quantitative complement to the t-SNE visualizations in Fig. 1–2.
>
> This makes the notion of “manifold connectedness” precise and measurable in our setting.
>
> ---
>
> ### **2. Why multi-view adaptive partitioning tends to “stitch” disconnected islands**
>
> Each MAPE view learns a few thresholds on the raw features (e.g., “age < 30”, “income > 1000”) and groups all nodes that satisfy the same combination of thresholds into one **cell**, assigning them a shared embedding.
>
> Now consider a simple auxiliary graph where we connect two nodes if they fall into the same cell in at least one view:
>
> - Even if two regions are far apart in the original feature space, they can become connected through a short chain of nodes that share cells across **different views**.
> - As we increase the number of views $K$, the probability that any two regions become connected by such chains **grows rapidly**.
>
> In **Appendix A.7**, we formalize this intuition. Under mild assumptions (each used cell contains a non-negligible fraction of data, and the views are reasonably diverse), the probability that two nodes share a cell in at least one view is bounded below by
>
> $$
> P(\text{share a cell in some view})
> \ge 1 - (1 - q_{\min})^K ,
> $$
>
> where $q_{\min}$ is a lower bound on the per-view co-assignment probability for semantically related nodes.
>
> Once this per-pair connection probability exceeds the standard connectivity threshold (on the order of $(\log M)/M$ for $M$ nodes), the “shared-cell” graph becomes connected with high probability.
> This yields a concrete mathematical condition under which multi-view adaptive partitioning reconstructs an approximately connected structure in the representation space.
>
> ---
>
> ### **3. Guarantees on the quality of the learned partitions**
>
> Beyond connectivity, we also analyze **how good** the learned partitions are.
>
> For each node pair $(i,j)$, we define a **co-assignment score**
>
> $$
> \hat\phi_{ij}
>   = \frac{1}{K}\sum_{k=1}^K
>     \mathbf{1}\{\text{nodes } i \text{ and } j \text{ fall into the same cell in view } k\}.
> $$
>
> Intuitively, if truly similar pairs have a slightly higher per-view probability of sharing a cell than dissimilar pairs, then:
>
> - as $K$ increases, a standard concentration argument shows that $\hat\phi_{ij}$ for similar pairs will, with high probability, exceed that for dissimilar pairs;
> - in other words, the empirical co-assignment matrix $(\hat\phi_{ij})$ becomes a **consistent estimator** of the underlying semantic similarity.
>
> This provides a guarantee that the learned partitions are not arbitrary: they capture meaningful semantic neighborhoods and support the reconstructed manifold structure.
>
> ---
>
> ### **4. Where the full theory appears in the revision**
>
> All formal definitions, assumptions, and proofs (including the exact statements corresponding to the above intuitions) are now provided in **Appendix A.7** of the revised manuscript.
> In the main text, we have also updated the discussion of *“restoring manifold connectedness”* to explicitly point to Appendix A.7, and these changes are highlighted in **orange**.

---

> ### Author Response · Authors · 2025-11-20
> **Response to W2 (Lack of detailed time/space complexity analysis)**
>
> We thank the reviewer for this comment. In the revised manuscript we add an appendix subsection ``Computational Complexity'' that formally analyzes both time and space costs of MAPGA. We first clarify that the shallow decision trees in Sec.~3.1 are used *only* to initialise the scalar split rules of MAPE; after initialisation the trees are discarded and are neither evaluated nor updated, so MAPGA is *not* a tree--GNN ensemble at training or inference time. The optional one-off initialisation step costs $O(K\, n\, N\, \log n)$ if implemented with CART, whereas one training epoch has complexity
>
> $$
> T_{\text{epoch}} = O(n\, K\, L_{\text{bar}})\+O(K\, m)\+O(L\, m\, d),
> $$
>
> which is linear in both the number of nodes $n$ and edges $m$, with $O(n\, K\, L_{\text{bar}})$ and $O(K\, m)$ explicitly quantifying the overhead of the $K$ parallel partition operators on top of a standard $O(L\, m\, d)$ GCN backbone. We further show that memory usage is also linear in $|V|$ and $|E|$ (sparse adjacency plus node embeddings), with a moderate codebook term governed by $K$, $L_{\text{bar}}$ and $d_0$. Full derivations are provided in the new appendix subsection.

---

> ### Author Response · Authors · 2025-11-20
> **Response to W3&Q2 (Limited analysis of decision-tree initialization impact)**
>
> We thank the reviewer for pointing out that the impact of the decision-tree initialization was not sufficiently analyzed. In our framework, the maximum tree depth $D$ is a key hyperparameter: it controls the **number and granularity of initial split rules** that define the adaptive partitions in each view.
>
> To systematically study how $D$ should be set and whether it affects final performance, we added an ablation study in the appendix (Figure 6, Appendix A.8). In this experiment, we sweep
> $D \in \{5,\dots,15\}$ on four representative datasets (Weibo, Amazon, YelpChi, Elliptic) and track both AUPRC and Rec@K.
>
> The main observations are:
>
> - **Stable performance for moderate depths.**
>   On **Weibo** and **Amazon**, both AUPRC and Rec@K curves are almost flat once $D \ge 6$, indicating that the model is robust to the exact depth choice in this range.
>
> - **Very deep trees can over-fragment simple features.**
>   On **YelpChi**, excessively large $D$ leads to a slight performance drop, consistent with over-fragmenting relatively low-dimensional features into many tiny regions.
>
> - **Deeper trees mildly help on complex, high-dimensional data.**
>   On **Elliptic**, which has high-dimensional transaction features, larger $D$ yields small but consistent gains, as additional splits allow the partitions to capture richer feature interactions.
>
> Overall, the variations across reasonable depths are **much smaller** than the gains of MAPGA over state-of-the-art baselines, suggesting that the method is not overly sensitive to the exact choice of $D$.
>
> In practice, we therefore fix $D$ to a **single shallow-to-moderate value** (e.g., $D=10$) within the empirically stable range and use the same depth for all experiments. The revised manuscript now (i) clarifies this design choice in the method section and (ii) explicitly refers to the new depth-ablation results in the appendix, with all related updates highlighted in **orange**.

---

> ### Author Response · Authors · 2025-11-20
> **Response to Q3 (Smooth surrogate for the Heaviside function and choice of α)**
>
> We thank the reviewer for raising this question. In Eq. (1) we approximate the Heaviside step function with a logistic surrogate:
>
> $$
> H(x) \approx \sigma_{\alpha}(x) = \frac{1}{1 + \exp(-\alpha x)} .
> $$
>
> where $\alpha$ is a temperature parameter controlling the steepness of the transition. During training, the binary decision $b_{l,m,k}$ is computed as
> $b_{l,m,k} = \sigma_{\alpha}\big( f^{(m)}_{n_l^k} - s_l^k \big)$
> so that gradients can be backpropagated to the split points $s_l^k$. At inference time we apply the corresponding hard threshold
> $H(x) = 1$ if $x \ge 0$ and $0$ otherwise.
>
> We treat $\alpha$ as a global hyperparameter shared by all views and rules. The current manuscript includes an ablation study in the supplementary material (Fig. 7), where we sweep $\alpha \in \{10, 20, \dots, 100\}$ and report AUPRC and Rec@K on four representative datasets (Weibo, Amazon, YelpChi, Elliptic). The curves exhibit a broad plateau: performance changes only slightly once $\alpha$ is moderately large, and the best results consistently occur for $\alpha$ in the range $40$--$60$. Guided by this study, we fix a single global value $\alpha = 50$ for all experiments in the paper, avoiding any per-dataset tuning while remaining in this stable regime. Section 3.1 and Appendix A.9 ("Effect of the surrogate slope $\alpha$") describe the surrogate and this choice of $\alpha$ in detail.

---

### Comment · Area_Chair_rUC4 · 2025-11-28

Dear Reviewers,

Thank you for your valuable time and expertise in reviewing this paper.

The authors have now submitted their rebuttal. We would appreciate it if you could review their responses and assess whether your concerns have been addressed, if you haven't done this.

Best regards,

AC

---

### Meta-Review · Area_Chair_NkTb · 2025-12-21

**Summary:**

The work is inspired by the difficulty of of GNNs in learning embeddings of nodes with tabular attributes in graph anomaly detection (GAD) and introduces a method called Multi-view Adaptive Partition Encoder (MAPE) to address this challenge. The key idea is to learn a set of subspaces through discretization and model global dependecies therein. The effectiveness of MAPE is evaluated on 10 GAD datasets, in comparison to 27 baselines.

**Reviewer Concerns:**

Concerns that have been addressed:
- All reviews point to unclear or insufficiently detailed method descriptions, including how tabular data is constructed as a graph, why GNNs are appropriate, how multiple views and partitions are instantiated, and how decision tree initialization affects performance. The rebuttal has provided sufficient details to clarify these concepts and/or items.


Concerns that have been partially addressed:
-  Two reviews highlight concerns about computational overhead, noting that generating many views and partitions may significantly increase time and memory costs. The rebuttal provides time complexity analysis of the proposed method, which helps understand the computational cost from a theoretical point of view, but no empirical computation cost is provided, making it difficult to directly evaluate the computation overhead.
- One reviewer note the absence of a formal theoretical explanation for why adaptive partitioning restores manifold connectedness, despite this being a central claim. The rebuttal clarifies how manifold connectedness is defined, measured, and theoretically guaranteed. However, this clarification is added as supplementary support, rather than as a major evidence in the main text; moreover, this theoretical analysis is a lengthy additional material (and probably new contributions), whose correctness and usefulness require additional comprehensive peer-review.
- The reviews identify a number of writing issues, repetition, and numerous typos. The rebuttal corrects the errors and typos. However, there are new format/typo issues in the updated manuscript.
- Major concerns are raised about the novelty of tabular node attribute modeling and long-range dependency handling, since they have already been studied in prior studies, particularly in GAAP, GADAM, and UniGAD. The rebuttal clarifies the differences between the proposed method and these prior methods. However, the AC tends to agree with the reviewers that the overlapping/similarity is too much, and the identified differences are not significant enough to make it to ICLR.

Concerns from the AC:
- Although the work makes claims in the area of graph anomaly detection, the model design is very generic, tackling a general node classification or supervised GNN problem. The AC finds that (i) the challenges discussed exist in almost all supervised GNNs on real-world graph datasets with tabular attributes, (ii) this applies to the long-range dependency issue too, and (iii) what are the unique challenges in graph anomaly detection targeted and how are they addressed in the model? The observations lead to a conclusion that the work is not properly positioned in terms of the problem formulation and model design.

**Reviewer Scores:**

There are three reviews, including two weak accepts (2x6s) and one reject (2). It is anticipated that one weak accept would retain his/her positive rating, while the another weak accept might drop to negative rating due to the concerns over time complexity analysis and the theoretical analysis discussed above. No change is expected to the reject rating due to the novelty concern and the absence of rigorousness in the writing.

---

### Decision · Program_Chairs · 2026-01-26

Reject